# The brain is required for normal muscle and nerve patterning during early *Xenopus* development

Celia Herrera-Rincon [1], Vaibhav P. Pai [1], Kristine M. Moran[1], Joan M. Lemire[1] & Michael Levin [1]

Possible roles of brain-derived signals in the regulation of embryogenesis are unknown. Here we use an amputation assay in *Xenopus laevis* to show that absence of brain alters subsequent muscle and peripheral nerve patterning during early development. The muscle phenotype can be rescued by an antagonist of muscarinic acetylcholine receptors. The observed defects occur at considerable distances from the head, suggesting that the brain provides long-range cues for other tissue systems during development. The presence of brain also protects embryos from otherwise-teratogenic agents. Overexpression of a hyperpolarization-activated cyclic nucleotide-gated ion channel rescues the muscle phenotype and the neural mispatterning that occur in brainless embryos, even when expressed far from the muscle or neural cells that mispattern. We identify a previously undescribed developmental role for the brain and reveal a non-local input into the control of early morphogenesis that is mediated by neurotransmitters and ion channel activity.

[1] Biology Department and Allen Discovery Center, Tufts University, 200 Boston Avenue, suite 4600, Medford, MA 02155-4243, USA. Correspondence and requests for materials should be addressed to M.L. (email: michael.levin@tufts.edu)

The brain and central nervous system (CNS) generate information that controls muscle activity to implement behavior in adult organisms. However, the CNS may also provide an instructive influence over the behavior of multiple cell types during the establishment, repair and maintenance of complex anatomical patterns in vivo. For example, the patterning disorder known as cancer has been linked to neural control[1]. Tumors are readily induced by denervation in cockroach[2], and are more readily induced by chemical carcinogens in denervated rabbit ears compared with contralateral controls bearing normal innervation[3]. The same has been observed with sarcomas implanted into normal or denervated frog limbs[4]. Even normal tissue can disorganize in the absence of neural signaling, as occurs in the papillae of the mammalian tongue after denervation[5]. Thus, the presence of growth control signals between the CNS and target cells has led to suggestions of the CNS as a potential approach to treat tumor progression in the clinic[6].

Fig caption content:
**a** Brain removal — Early — Late (Ctrl / BR−)

**b** sm, hb, e, fb, cg

Correct / Incorrect

Early — Late

**c** Control (Ctrl) **f**

**d** Brainless (BR−) **g**

**e** Rostral–caudal axis
** ** **
90° 135° 180°
Mean angle of myotome fibers at early

**h** Rostral–caudal axis
Ctrl / BR−
* * *
90° 135° 180°
Mean angle of myotome fibers at late

Likewise, pattern regulation during regeneration requires neural signals. Appendage regeneration has long been known to be dependent on local neural supply[7], although this is apparently an acquired addiction to developmental presence of innervation[8]. Recent studies have implicated roles for neural signaling in bone regeneration[9] and epidermal de-differentiation[10]. The role of CNS in regeneration is instructive, not merely permissive. The polarity and contiguity of the CNS determines head/tail specification in a number of invertebrate models, including planaria[11, 12]. In salamander, the presence of nerve in a non-native location can induce the formation of ectopic limbs at wound sites[13]. Moreover, in the frog model, focal damage to the spinal cord induces alterations in the specific pattern of tail regeneration, with distinct morphological changes resulting from spinal cord interruptions at different locations[14]. Despite recent advances[15], the molecular connection and reciprocal influence between nerves and patterning remains poorly understood[16], especially with respect to the ultimate source of neutrally mediated patterning signals.

One of the most interesting contexts for neural pattern control is embryogenesis, when anatomical structures are first established. Parasympathetic innervation controls morphogenesis of the submandibular gland, influencing the branching and general patterning of the organ[17]. Muscle cells are also affected by interactions with nerves[18], especially in terms of gene expression of myogenic regularity factors (MRFs)[19]. Complementing molecular and cell-level readouts, regenerative medicine approaches to birth defects and bioengineered organs requires addressing neural inputs into large-scale patterning. However, possible roles of the brain (or brain-derived signals) for early patterning, long before behavior begins, remain unknown.

In *Xenopus*, somitogenesis occurs in an anterior-posterior, bilaterally symmetric manner. In the clock and wavefront model[20, 21], the presomitic mesodermal cells dynamically oscillate between permissive and nonpermissive conditions for segmentation (clock) as the embryonic development proceeds from anterior to posterior direction (wavefront along with tissue shortening-mediated Doppler effect) resulting in formation of segmented somites. Wnt, Shh, BMP, Fgf and Notch pathways are involved in regulating the clock with rhythmic activation of these pathways conserved across vertebrates. The intersection of antagonistic gradients of retinoic acid (anterior to posterior) and Fgf and Wnt (posterior to anterior) determine the wavefront[22, 23]. Given the tandem physiology of nerves and muscles, might innervation help orchestrate the symphony of signals that leads to somitogenesis? Does the presumptive nervous system release instructive signals that might be involved in somitogenesis and muscle development? Is there any element of this process co-opted into muscle regeneration? Similarly, the peripheral innervation forms a stereotypic and complex neural network,

which also must be precisely patterned in a way that integrates size and positional information across the whole body[24].

Most of the existing knowledge about neural inputs focuses on the local microenvironment; very few studies have examined the contribution of the CNS in providing long-range instructive cues to muscle or peripheral innervation. To probe the role of distal neural structures and to search for tissues that could provide complex nerve-mediated information to downstream patterning targets, we focused on the brain. Here, we establish a brainless *Xenopus* embryonic model to reveal a role for the brain in the patterning of muscle (somite) and peripheral innervation. Crucially, we show that defects induced by extirpation of the brain, or by teratogenic agents in the context of normal embryogenesis, can be largely rescued in the absence of brain by the dorsal overexpression of an exogenous ion channel or exposure to a muscarinic-receptor antagonist. The characterization of defects and their repair by global (pharmacological) or spatially targeted (molecular-genetic) reagents reveal the first molecular components of the signaling cascade by which the brain directs complex embryonic patterning processes.

## Results

**Brain is required for normal muscle development and patterning.** To investigate the possible role of brain-derived signals for muscle morphogenesis and patterning, we established an assay in *Xenopus laevis*: brain removal at stage 25 (Fig. 1a, b), performed at a time when its main subdivisions (forebrain, midbrain and hindbrain) and the rostral-caudal and dorsal-ventral axis are already defined[25]. By removing the brain at the early tailbud stage (i.e., when somitogenesis is starting), we were able to study muscle structure development in completely brainless developing animals (> 85% of the microsurgeries we performed resulted in viable animals). To determine whether the brain is required for the onset and/or patterning of myotomes, we evaluated the muscle phenotype at two relevant time points, corresponding to the different myogenic waves (reviewed in ref. [23]): early- (stages 30–41; first and second waves completed) and late- (stages 42–48; third wave completed) stages after brain removal, respectively.

Soon after brain removal (stages 30–41), animals developing without a brain (BR⁻) began to display a notable decrease (−43 ± 7%) in the collagen density of the myotomes, compared to the control animals (Ctrl) (OD mean value of 64 ± 8 units for BR⁻ group compared to 113 ± 13 units for Ctrl group; *t*-test $P < 0.01$; $n = 79$) (Fig. 1c, d, *turquoise* and *magenta short arrows*). Analysis of the muscle structure revealed that the somites were 25% shorter in BR⁻ than those found in control animals (101 ± 24 μm vs. 148 ± 13 μm, *t*-test $P < 0.01$; $n = 75$) (Fig. 1c, d, *double-headed arrow* indicates the length of one myotome). Moreover, the spatial organization of the somatic muscle was also

**Fig. 1** The absence of the early brain leads to abnormal muscle development and patterning. **a** After fertilization, the brain was removed from stage 25 embryos to generate BR⁻ animals. Morphological evaluation of muscle phenotype was performed at early- (stages 30–41) and late- (42–48) stages. **b** Lateral views of stage 25 embryos before (*left*) and after (*right*) brain removal. The area occupied by the developing brain is marked with a *white-dashed line*. (*left*) rostral is *left* and dorsal is *up*. Scale bar, 250 μm. cg, cement gland; e, eye; fb, forebrain, hb, hindbrain, sm, somites. **c–h** The brain is required for normal muscle development and patterning, as shown after quantitative evaluation of collagen density (*short arrows*), length of myotome fibers (*double-headed arrows*), central body axis and myotome angle (*overlaid dashed* axis and *arrowhead*-like lines) at early **c**, **d** and late **f–h** stages. At the onset of development, BR⁻ embryos possessed a lower collagen density in myotome fibers (*magenta arrow* in **c** compared to *turquoise arrow* in **b**), a significantly more open central angle along the rostra-caudal axis **e** and shorter somites than control (Ctrl) embryos. During development, defects in the organization of central body axis and muscle patterning were not corrected at any anatomical level in BR⁻ (*magenta dashed lines* in **g** compared to *turquoise dashed lines* in **f**. The mean angle for BR⁻ is significantly displaced to 180°, compared to those in Ctrl (H). **c**, **d**, **f**, **g** Photomicrographs taken under polarized light. Rostral is *upper right* and dorsal is *up*. Turquoise and *magenta arrows* indicate correct and incorrect anatomical pattern, respectively. *Scale bar*, 500 μm. **e**, **h** Graphic representation of the mean angle of myotome fibers at rostral, central and caudal levels (*blue squares*) of Ctrl (*white*) and BR⁻ (*gray*) embryos. Data represent the mean and s.d. of three independent replicates ($n = 75$ animals per group). P values after *t* (equal variances, *black* labels) or Mann–Whitney (unequal variances, *blue* labels) tests are indicated as **$P < 0.01$, *$P < 0.05$, ns no significant difference

**Table 1 Mean angle of the muscle fibers along the rostro-caudal axis in Ctrl and BR⁻ embryos**

|  | Early stage | | Late stage | |
|---|---|---|---|---|
|  | **Ctrl** | **BR⁻** | **Ctrl** | **BR⁻** |
| Rostral | 132 ± 15° | 148 ± 13°** | 123 ± 12° | 133 ± 20°* |
| Central | 115 ± 12° | 134 ± 16°** | 114 ± 10° | 123 ± 18°* |
| Caudal | 110 ± 10° | 122 ± 12°** | 111 ± 11° | 118 ± 15°* |

Ctrl: control; BR⁻: brainless

Mean angle values for Ctrl and BR⁻ groups are given for the rostral, central and caudal level of the embryo body, respectively, at both early- and late-stages. Values are presented as mean angle ± s.d. Statistically significant intragroup differences after unpaired and two-tailed Student's *t*-test are highlighted by *$P < 0.05$, **$P < 0.01$ (blue labels are for $P$ values after two-tailed Mann–Whitney test)

**Table 2 Proportion of individuals with aberrant phenotype in Ctrl and BR⁻ populations ($p_i/n_i$) after drug treatment**

|  | Early Stage | | Late Stage | |
|---|---|---|---|---|
|  | **Ctrl** | **BR⁻** | **Ctrl** | **BR⁻** |
| No drug | 30/220 | 80/120** | 38/228 | 132/164** |
| Scopolamine | 6/41 | 24/56** | 6/38 | 9/37 ns |
| Carbachol | 14/40 | 45/60** | 29/61 | 45/47** |

Ctrl: control; BR⁻: brainless; ns: not significant

Number of aberrant embryos ($p_i$) and the pooled-sample size ($n_i$) is given for untreated (no drug), scopolamine- and carbachol-treated Ctrl and BR⁻ populations, respectively, at both early- and late-stages. Statistically significant intragroup differences after *z*-test are highlighted by *$P < 0.05$, **$P < 0.01$, ns $P > 0.05$

perturbed. The body axis and mean angle of the muscle fibers along the anteroposterior axis were significantly altered in BR⁻, with a mean of $16 ± 2°$ more-opened angles compared to Ctrl ones (*t*-test $P < 0.01$ for rostral and caudal levels, Mann–Whitney test $P < 0.01$ for central level levels; $n = 75$) (Fig. 1e and Table 1).

We then asked whether the early defects in muscle patterning were also present during subsequent development in BR⁻ embryos. To address this question, we analyzed the number of somites (indicator of segmentation) and the fine muscle structure (angle and length of the myotome fibers) at late stages (stages 42–48). No significant differences were detected for the mean number of somites between Ctrl and BR⁻ embryos ($33 ± 3$ vs. $32 ± 4$; *t*-test $P = 0.74$; $n = 75$), suggesting that the brain was not functionally implicated in segmentation *per se*, and that our assay does not generally (nonspecifically) impair embryogenesis of the somites. In contrast, the analysis of the fine muscle structure confirmed that the defects in myotome organization were not repaired during subsequent development. Myotomes in late-staged BR⁻ embryos were 10% shorter ($136 ± 14 \mu m$ vs. $160 ± 13 \mu m$, *t*-test $P < 0.01$; $n = 75$) than in control animals. Likewise, the central axis in the late-stage BR⁻ was significantly displaced, somites lacked the typical chevron-shape, and a difference of $-9 ± 1°$ in the mean angle of the myotome fibers was detected when compared to Ctrl animals (Mann–Whitney test $P < 0.05$ for rostral and central levels, *t*-test $P < 0.05$ for caudal level; $n = 75$) (Fig. 1f–h, overlaid dashed axis and arrowhead-like lines, and Table 1). Control animals subjected to sham surgeries where either yolk mass or tailbud was resected (Yolk⁻ or Tail⁻ embryos, respectively) displayed normal muscle architecture, indistinguishable from control myotome fibers, both in terms of length/definition and angle of the central myotomes (at distance from the initial resection, in the case of Tail⁻ sham-embryos) (Supplementary Fig. 1A–G), demonstrating that surgery per se (and subsequent regenerative responses) do not induce muscle mispatterning. We conclude from these data that while the brain is not required for the somite segmentation (partitioning of the presomitic mesoderm into somites), it has a key role in both the onset and establishment of correct muscle patterning and structure.

Muscle organization was adversely affected both at the microscopic tissue organization level, as well as macroscopically – at the animal morphological level (incidence of aberrant phenotype within both Ctrl and BR⁻ population; Table 2). A macroscopic evaluation of embryo morphology revealed a higher percentage of abnormal embryos in the BR⁻ population, both at early ($66 ± 6\%$ vs. $13 ± 1\%$; *z*-test, $P < 0.01$) and at later stages ($80 ± 1\%$ vs. $16 ± 2\%$; *z*-test $P < 0.01$) compared to Ctrl animals. Muscle organization was affected by brain absence rather than any indirect detrimental effect of tissue removal surgery, as demonstrated the percent of aberration in Yolk⁻ population

($14 ± 2\%$ and $15 ± 5\%$ at the onset and later in development, respectively; *z*-test compared to Ctrl $P > 0.05$ for both cases; Supplementary Fig. 1H).

Taken together, these results clearly indicate the importance of the early embryonic brain for normal development of muscle structure occurring at considerable distance.

**Scopolamine exposure rescues the BR⁻ muscle phenotype**. To characterize the brain-dependent signals that were necessary for normal muscle development, we first asked if the aberrant BR⁻ muscle phenotype could be rescued by pharmacological treatment. Neurotransmitters, such as acetylcholine, are conserved and ubiquitous mediators of the brain's electrical activity on other organs and tissues in the body. We reasoned that similar mechanisms might be at work before behavior, in the developmental process, as neurotransmitters not only mediate adult physiological function downstream of bioelectrical events but also play a developmental role in the patterning and formation of the synapses that they subserve[26]. Therefore, we targeted this pathway to attempt to understand and recapitulate the brain's role in embryogenesis. We tested several cholinergic drugs that were known to target muscarinic (mAChRs) and nicotinic cholinergic receptors (nAChRs) and are standard tools for altering brain performance, especially in terms of memory, attention, and relevant-stimulus processing[27].

Ctrl and BR⁻ embryos were treated with scopolamine (a muscarinic-receptor antagonist)[28] or carbachol (a dual muscarinic- and nicotinic-receptor agonist)[29] (Fig. 2a–h). When comparing percent of abnormal embryos (macroscopic phenotype) within BR⁻ and Ctrl populations for each pharmacological treatment, we found that the absence of brain leads to a higher incidence of abnormalities at the onset of the development, irrespective of the drug used (Fig. 2a and Table 2). When intergroup analysis is performed, however, significant differences between scopolamine-treated BR⁻ and untreated BR⁻ were detected ($38 ± 18\%$ vs. $66 ± 8\%$; *z*-test $P < 0.01$), indicating that scopolamine could have a protective effect on the overall morphology when the brain is absent. Our analysis at late stage (Fig. 2b and Table 2) revealed that, while carbachol treatment increased the incidence of aberration in the BR⁻ population (reaching almost 100% of embryos with abnormalities), scopolamine treatment abrogated those effects completely. Consequently, no significant differences were found between scopolamine-treated BR⁻ and scopolamine-treated Ctrl ($25 ± 9\%$ vs. $15 ± 5\%$; *z*-test $P = 0.36$), though significant differences were observed between scopolamine-treated BR⁻ and untreated BR⁻ ($25 ± 9\%$ vs. $80 ± 2\%$; *z*-test $P < 0.01$).

We then microscopically analyzed the structure of the myotomes in the different drug-treated groups (Fig. 2c–f for comparative micrographs and Fig. 2g, h for statistical

comparisons). Evaluation of somitogenesis and fine somatic muscle structure revealed that both the organization and size of myotomes in BR⁻ treated with scopolamine resembled the typical Ctrl muscle phenotype (*turquoise arrows* in Fig. 2c, e), and differed clearly from the typical BR⁻ muscle phenotype

(represented in Fig. 2d). Both the mean number of somites (32 ± 2) and length of muscle fibers at early- (141 ± 7 μm) and late- (157 ± 17 μm) staged scopolamine-treated BR⁻ were statistically similar to the values for the untreated Ctrl population (with mean number of 32 ± 2 somites and mean lengths of

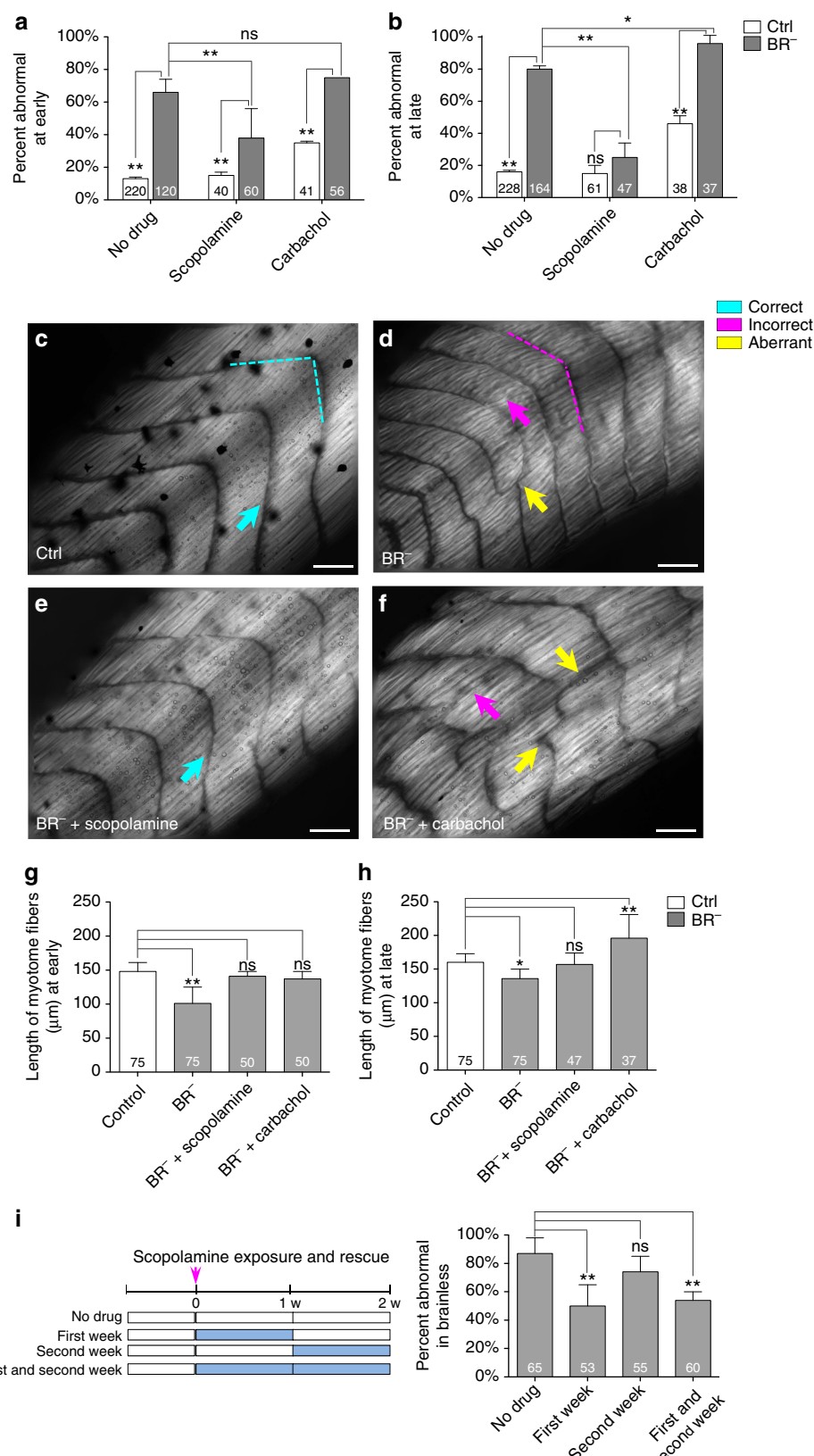

$148 \pm 13\,\mu m$ at early stage and mean lengths of $160 \pm 13\,\mu m$ at late stage; Fig. 2g, h). Conversely, carbachol treatment in BR⁻ had dramatic negative consequences for muscle formation and patterning, which was especially clear at the later stages, deviating the organization of somites and myotomes from the typical BR⁻-induced aberrant muscle phenotype. At early stages, the size of the myotomes (mean length of $137 \pm 11\,\mu m$) was similar to what was measured in the Ctrl embryos. However, later on in development, a significantly lower number of somites ($24 \pm 4$; $P < 0.01$ after post-hoc Bonferroni's test), longer myotome fibers ($196 \pm 35\,\mu m$; $P < 0.01$ after post-hoc Dunn's test), and complete asymmetric and disorganized muscle patterning were observed in the carbachol-treated BR⁻ (*magenta* and *yellow arrows* in Fig. 2f). Taken together, we conclude that the brain may inhibit the muscarinic pathway to achieve correct organization of the somatic muscle system and that the absence or prevention of muscarinic signaling (for example, the pharmacological antagonism mediated by scopolamine) is able to rescue the aberrant BR⁻ muscle phenotype.

We next sought to determine when during muscle development the rescue effect of scopolamine occurs. BR⁻ embryos were exposed to scopolamine for different lengths of time (Fig. 2i *left*, for a schematic representation of drug-exposure timing). We observed that scopolamine treatment is able to fix the BR⁻ muscle phenotype if the drug exposure starts immediately after the brain removal (Fig. 2i, *right*), regardless of whether BR⁻ are treated for one (first-week experimental group) or 2 weeks (first- and second-week experimental group). In either case, treatment was able to significantly decrease the percentage of abnormal tadpoles ($51 \pm 13\%$ and $55 \pm 7\%$, respectively; $z$-test $P < 0.01$ for both groups) when compared to the BR⁻ population that had not been exposed to drug ($87 \pm 11\%$ of abnormalities). Our results indicate that scopolamine acts on events during the first week (from stages 25 to 37) after brain removal.

**Ectopic HCN2 expression rescues BR⁻ muscle phenotype.** Considering the many examples of muscle formation and patterning mediated by bioelectrical signaling in both mammals[30] and amphibians[31], as well as the ability of some channels to rescue profound embryonic defects[32], we wondered whether the exogenous expression of a specific ion channel, hyperpolarization-activated cyclic nucleotide-gated ion channel 2 (HCN2), could counter the effects of brain removal. The HCN2 channel is known to be an important modulator of functional bioelectric state[33] and can hyperpolarize cells, and it has been recently shown to be implied in firing and rhythmic properties of the cholinergic neurons in both CNS and gastrointestinal tract[34, 35]. We found the HCN2 channel to be endogenously

expressed in the developing neural tube along the base and lateral regions and in the perisomitic area (Supplementary Fig. 2). To address the role of bioelectric signaling during muscle development in organisms incapable of relaying signals from their brains to other tissue, embryos were injected at two-cell stage, in both blastomeres, with mRNA encoding wild-type HCN2 (Fig. 3a, turquoise arrows, HCN2-WT group). Uninjected and water-injected embryos served as controls. In addition, in order to understand the signaling between HCN2-expressing cells and the structural muscle outcome (either a local or long-distance effect), we evaluated the muscle phenotype after injection of HCN2-WT mRNA in only one side (left-right) of the embryo (1/2 HCN2-WT group). Co-injection with lacZ mRNA (as a reporter of injected cells' progeny; Fig. 3b, *left*, *blue arrow*) and detection of the β-galactosidase (β-gal) distribution were used to select and evaluate quantitatively those embryos with strongly unilateral expression (Fig. 3b, *right*, *blue arrow* indicates high β-gal expression and, thus, ipsilateral injected side).

First, we quantified the proportion of embryos with abnormal phenotype within each BR⁻ population (untreated or uninjected, water-injected, HCN2-WT and 1/2 HCN2-WT; Fig. 3c). Our results revealed that the expression of a WT HCN2 mRNA in BR⁻ clearly reduced the onset of abnormalities in the macroscopic morphology and significantly decreased the percent of abnormal embryos within the population (from $85 \pm 7\%$ in non-injected BR⁻ embryos to $45 \pm 7\%$ in HCN2-WT BR⁻ group; $z$-test $P < 0.01$). We observed the same rescue effect of HCN2 channel when only one side of the animal was injected. The incidence of aberrant phenotypes in the 1/2 HCN2-WT BR⁻ group was significantly lower than that for the regular (or uninjected) BR⁻ population ($65 \pm 3\%$ of aberrant individuals in the 1/2 HCN2-WT BR⁻ group; $z$-test $P < 0.01$ compared to the non-injected BR⁻ group).

We then microscopically analyzed the somatic-myotome patterning in BR⁻ embryos under the different ion channel misexpression conditions (Fig. 3d–g). Analysis of fine muscle structure revealed that the HCN2-WT BR⁻ mutants exhibited somites and myotomes that were perfectly organized, differing clearly from the typical BR⁻ muscle phenotype (Fig. 3d, e for comparative micrographs). The differences in the number of somites ($30 \pm 2$) and the mean length of myofibers ($134 \pm 11\,\mu m$ at early stage and $165 \pm 15\,\mu m$ at late stage) were not statistically significant from those measured in uninjected Ctrl embryos (after post-hoc Bonferroni's test). The unilateral HCN2 expression in animals developing without brain had a protective effect on the muscle organization of both the local side and the uninjected contralateral side. Like the results for the both sides HCN2-injections, the size of myotome fibers in the uninjected contralateral side were similar to the Ctrl embryos, both at the

**Fig. 2** Scopolamine rescues the BR⁻ muscle phenotype. **a**, **b** Quantification of the mean percentage of abnormal embryos and statistical comparisons among Ctrl and BR⁻ populations under normal conditions and after drug treatment, at early- (**a**) and late- (**b**) stages after brain removal. Values are plotted as mean % ± s.d. (no-pooled data from, at least, three different replicates). **c–f**. Typical muscle phenotype for Ctrl (**c**) and BR⁻ (**d**), and BR⁻ after scopolamine (**e**) or carbachol treatment (**f**), as seen under polarized light. Rostral is *upper right* and dorsal is *up*. Turquoise, magenta and yellow arrows indicate correct, incorrect and aberrant formation, respectively. *Scale bar*, 100 μm. **g**, **h** Quantification of the mean length of myotome fibers and statistical comparisons among untreated Ctrl and untreated BR⁻, scopolamine-treated BR⁻ and carbachol-treated BR⁻ at early- (**g**, one-way ANOVA, $P < 0.01$) and late- (**h**, Kruskal–Wallis test, $P < 0.01$) stages after brain removal. No significant differences after a posteriori analysis were detected among the different Ctrl groups. Data represent the mean and s.d. of three independent replicates. **i**. Scopolamine exposure and rescue effects on BR⁻ phenotype. (*left*) Graphical representation of the different exposure times to scopolamine in BR⁻, after brain removal ($t = 0$, magenta arrow; white band means no drug and blue bands means drug treatment) and for a 2-week (2w) period. First-week experimental group was exposed to scopolamine immediately after brain removal and consecutively for the first week. Second-week animals were exposed to the drug 1 week after the brain removal, for 1-week period. First- and second-week animals were exposed to scopolamine immediately after the brain removal and for the 2 next consecutive weeks. (*right*) Quantification of the mean percentage of embryos with abnormal phenotype within each BR⁻ group. Values are plotted as mean % ± s.d. (no-pooled data from three different replicates). For all panels, number in bars indicates *n* or number of embryos analyzed for each group. *P* values after *z*-test **a**, **b**, **i** and post-hoc Bonferroni's **g** or Dunn's test **h** are indicated as **$P < 0.01$, *$P < 0.05$, ns $P > 0.05$

beginning (early stage: $139 \pm 13\,\mu m$) and during the development course (late stage: $170 \pm 14\,\mu m$). Taken together, these results indicate that the ectopic expression of the HCN2 channel counteracts the effects of a missing brain during the somitic myogenesis, even when the ectopic HCN2 channel is present on the other side of the animal (a strongly non-local effect).

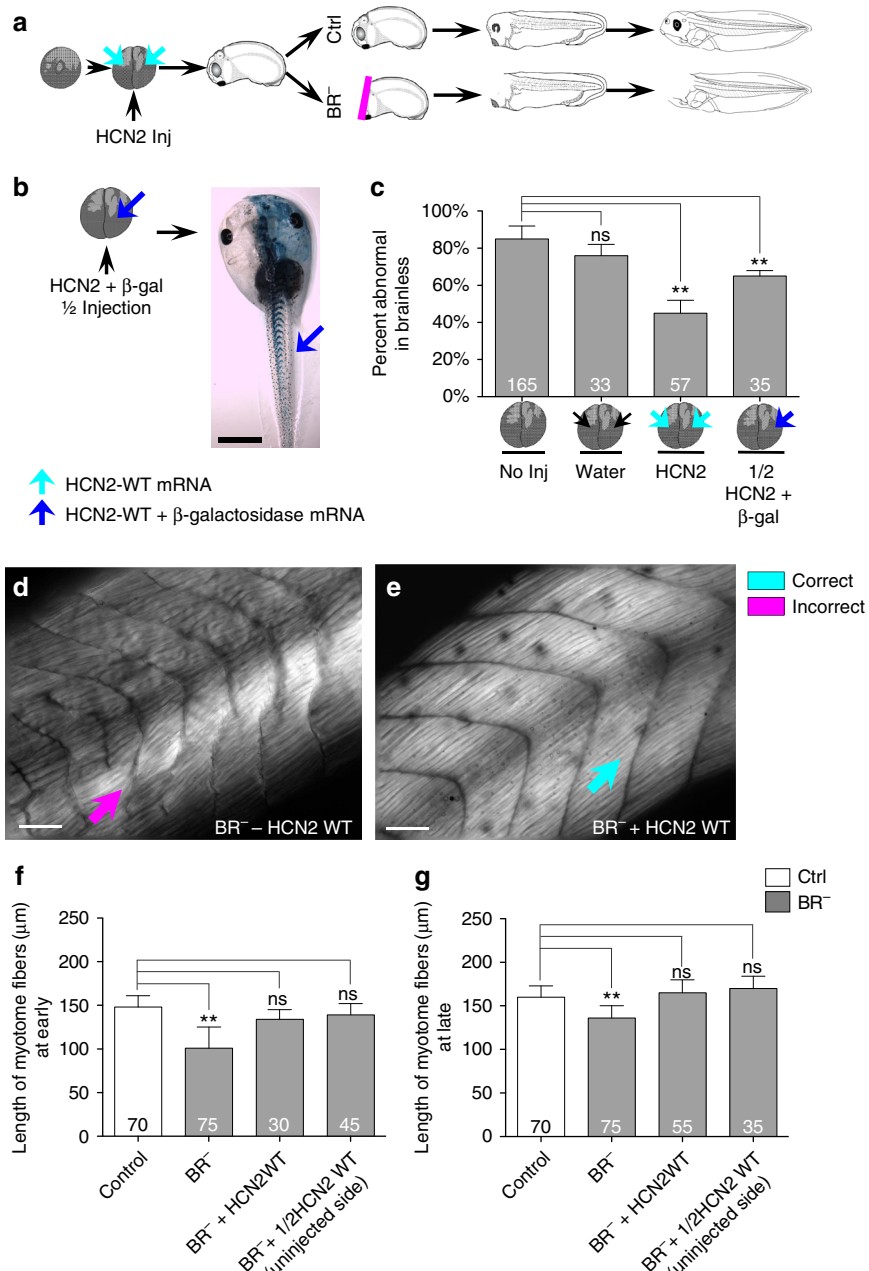

**Fig. 3** Ectopic expression of HCN2 rescues the BR⁻ muscle phenotype. **a** Embryos were microinjected (Inj) with HCN2 mRNA (wild-type channel, WT) either in the two cells (HCN2 WT-group, *turquoise arrows*) or in one cell (1/2 HCN2 WT, see **b** *blue arrow*) at the two-cell stage. Brain was removed at stage 25, and animals with and without brain (Ctrl and BR⁻, respectively) were analyzed for muscle structure and patterning at early- (stages 30–41) and late-stage (stages 42–48). **b** 1/2 HCN2-WT injection: embryos were microinjected with HCN2 and lacZ mRNA in one of the cells at two-cell stage (*blue arrow*). The injection side was confirmed by enzymatic detection of β-galactosidase, β-gal (dorsal view of one Ctrl animal is showed on the right). Rostral is up. *Scale bar*, 1 mm. **c** Quantification of the mean percentage of abnormal embryos (macroscopic phenotype) and statistical comparisons between uninjected BR⁻ embryos (No Inj) and the different injected-BR⁻ populations (Water, *black arrows*: water-injection in the two cells; HCN2: HCN2-WT mRNA injection in the two cells; 1/2 HCN2 + β-gal: co-injection of HCN2-WT and lacZ gene reporter in one LR side). Values are plotted as mean % $\pm$ s.d. (no-pooled data from two different replicates). **d**, **e** Typical muscle phenotype for uninjected BR⁻ (**d**) and HCN2-WT injected BR⁻ (**e**), as seen under polarized light. Rostral is *upper right* and dorsal is *up*. *Scale bar*, 100 μm. **f**, **g** Quantification of the mean length of myotome fibers and statistical comparisons among uninjected Ctrl and uninjected BR⁻ (BR⁻), HCN2-WT injected BR⁻ and 1/2 HCN2-WT (measured on uninjected contralateral side) at early- (**f** one-way ANOVA, $P < 0.01$) and late- (**g** one-way ANOVA, $P < 0.01$) stages after brain removal. No significant differences after a posteriori analysis were detected among the different Ctrl groups. Data represent the mean and s.d. of two independent replicates. For all panels, number in bars indicates n or number of animals for each group. $P$ values after after $z$-test **c** or post-hoc Bonferroni's test **f**, **g** are indicated as **$P < 0.01$, ns $P > 0.05$

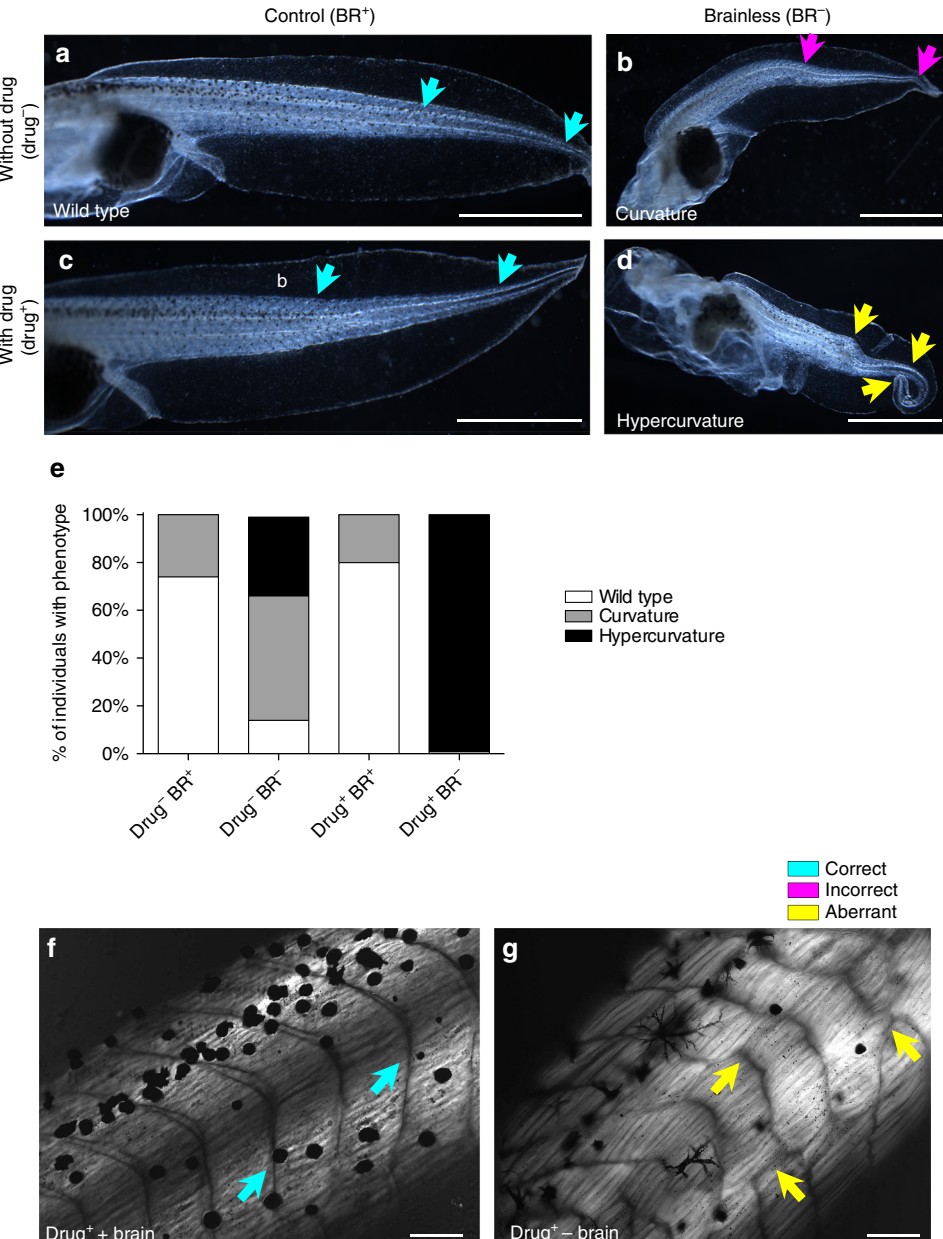

**Fig. 4** The brain can prevent drug-induced abnormalities of body patterning from occurring. **a–d** Lateral view of stage-45 tadpoles with brain (*left column*; Control, BR$^+$) or without brain (*right column*; Brainless, BR$^-$) after housing in normal conditions (top row; no drug treatment, Drug$^-$) and after continuous treatment with 10 μm (*RS*)-(Tetrazol-5-yl)glycine (RS, an NMDA receptor agonist), respectively. Drug treatment in Control animals (**c**, Drug$^+$ BR$^+$) did not produce alterations in tail patterning (*turquoise arrows* in **c** similar to **a**), and there was no incidence of aberrant or hypercurved phenotypes. Drug treatment in BR$^-$ animals (**d**, Drug$^+$ BR$^-$) lead to a completely aberrant population, with highly curved phenotypes (different to those in BR$^-$ without drug treatment, *yellow arrows* in **d** compared to *magenta arrows* in **b**). Rostral is to the *left* and dorsal is *up*. Turquoise, *magenta* and *yellow arrows* indicate correct, incorrect and aberrant tail modules, respectively. *Scale bar*, 1 mm. **e** Analysis of the phenotype distributions within each experimental group showed that RS is able to induce a significantly aberrant body patterning (a 'hypercurvature' phenotype) only if the brain is absent. Data represent the pooled distribution of three replicates ($n = 75$ animals per group). $P < 0.01$ for $X^2_{(0.05,\ 6)}$. **f, g** Evaluation under polarized light of drug-treated animals, with brain (**f**) and without brain (**g**), revealing clear muscle defects, both in structure and overall patterning, when the brain is not present (*yellow arrows* in g). This disorganization is not present in drug-treated control animals, exhibiting normal somites and myotome fibers (*turquoise arrows* in **f**, see Fig. 2c for similarity to Ctrl group). *Turquoise, magenta* and *yellow arrows* indicate correct, incorrect and aberrant muscle structure, respectively. *Scale bar*, 100 μm

**Brains protect tail patterning from effects of teratogens.** Having seen that the brain provides a beneficial patterning influence for aspects of development, we sought to test possible interactions of this effect with teratogenic agents. Could the adverse effects induced by drugs on macroscopic phenotype and tail patterning be prevented if the brain were present? Teratogenic agents targeting the GABAergic, glutamatergic, adrenergic and dopaminergic pathways have been shown to perturb pre-nervous functions of ion channels and neurotransmitter receptors[36, 37]. Thus, we performed a loss- and gain-of-function screen on our BR$^-$ animals (brain removal at stage 25, followed by immediate-drug exposure) with different neuroactive agents, in order to identify drugs that provoked the most severe phenotype in the BR$^-$ population (i.e., those that induce more highly aberrant

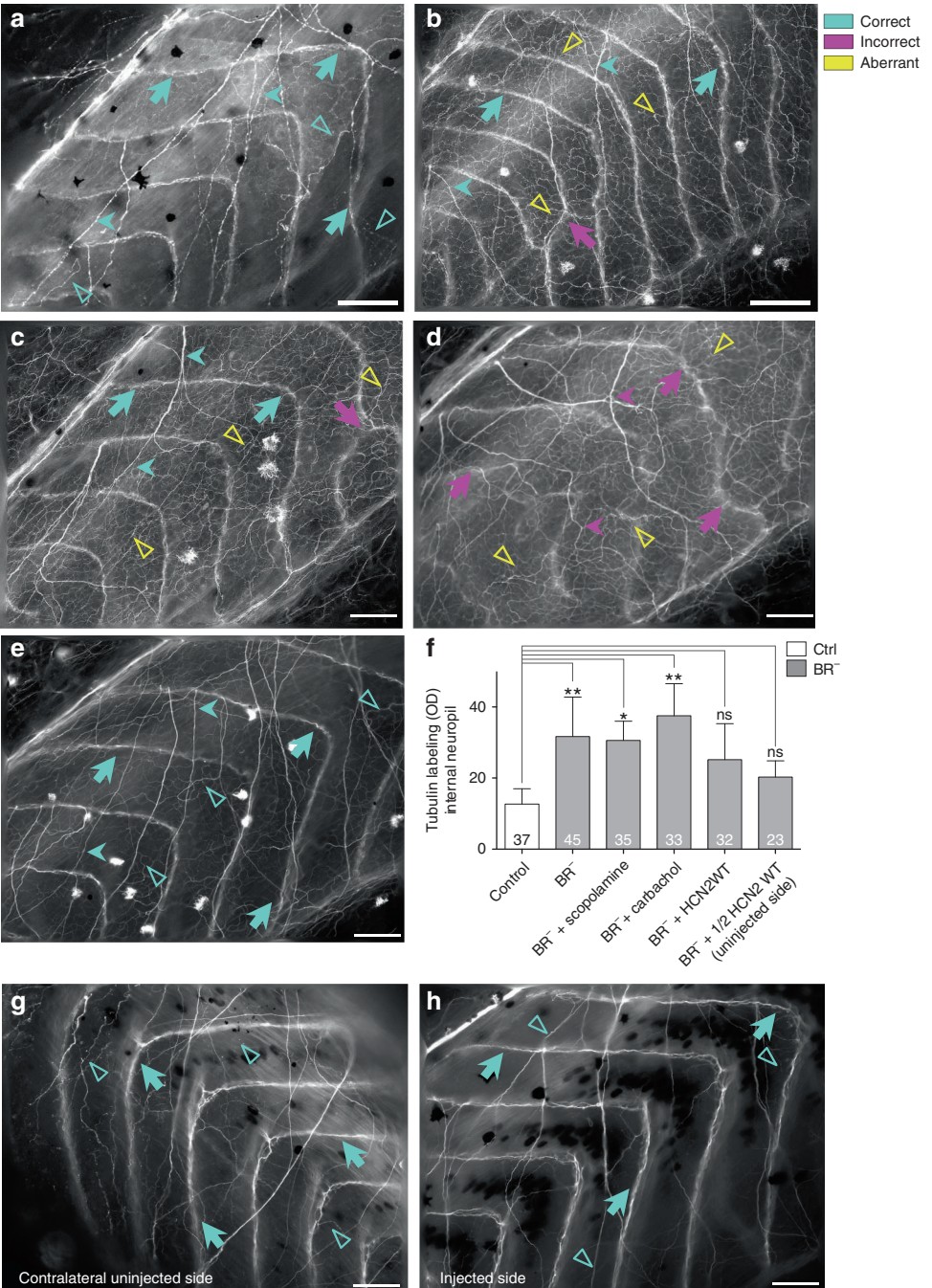

**Fig. 5** The absence of a brain generates an abnormal neural network in the entire animal body. **a**, **b** Acetylated-tubulin (Tub) immunoexpression for Ctrl (**a**) and BR⁻ (**b**) animals. There three types of nerve fibers: (i) commissural fibers (dorsoventral axis, *long arrows*); (ii) longitudinal fibers (anteroposterior axis, *short arrow*); and (iii) internal neuropil (no defined axis, *unfilled triangles*). Animals developed without a brain show normal commissural and longitudinal nerve fibers (*turquoise long arrows* in **b**), with some alterations (*magenta long arrow*), but a dense internal neuropil (*yellow unfilled triangles*). **c**, **d** Tub-immunoexpression for BR⁻ treated with cholinergic drugs: scopolamine (**c**) and carbachol (**d**). Scopolamine treatment was not able to rescue the aberrant internal network (*yellow unfilled triangles* in **c**), and carbachol-treated animals exhibited a chaotic nerve patterning (*magenta* and *yellow arrows* in **d**). **e** Ectopic HCN2-WT expression (injected in both cells at two-cell stage) fixed the BR⁻–induced internal nerve branching. **f** Quantification of the mean OD of internal neuropil and statistical comparisons among untreated/uninjected Ctrl and untreated/uninjecetd BR⁻ (BR⁻, without drug treatment nor ion channel misexpression), scopolamine-treated BR⁻ (BR⁻ + scopolamine), carbachol-treated BR⁻ (BR⁻ + carbachol), HCN2-WT both-sides injected BR⁻ (BR⁻ + HCN2 WT), and HCN2-WT LR side-injected BR⁻ (BR⁻ + 1/2 HCN2 WT) embryos (one-way ANOVA, P < 0.01). No significant differences after a posteriori analysis were detected among the different Ctrl groups. Data represent the mean OD units and s.d. of two independent replicates. Number in bars indicates *n* or number of animals analyzed for each group. P values after post-hoc Bonferroni's test are indicated as **P < 0.01, *P < 0.05, ns P > 0.05. **g**, **h**. Ectopic HCN2 expression in only one LR side fixes the BR⁻-induced internal nerve branching. Tub-immunoexpression on β-gal-reacted sections (dark deposits) in a 1/2 HCN2-WT BR⁻, showing both the contralateral uninjected side (**g**) and the injected (**h**) of the same embryo. Aberrant neural network was completely rescued (*turquoise arrows*), exhibiting a similar nerve pattern to the Ctrl group in both sides. **a**-**e**, **h**: Rostral is *upper right* and dorsal is *up*. **g**: Rostral is *upper left* and dorsal is *up*. *Scale bar*, 100 µm

phenotypes than the ones induced by brain removal). We observed that introduction of (RS)-(Tetrazol-5-yl)glycine (RS), an agonist of the NMDA-glutamate receptor[38] significantly increased the occurrence of aberrant tail phenotypes within BR⁻ embryos ($P < 0.01$ for $X^2_{(0.05,6)}$; Fig. 4) after 2 weeks of treatment (stages 42–48).

The most frequent tail phenotype in (untreated) BR⁻ embryos is characterized by a single lateral bending, starting approximately in the one-third posterior of the tail (see the most anterior arrow drawn on Fig. 4a–d photomicrographs). Continuous treatment with RS in BR⁻ caused a $99 \pm 3\%$ of highly bent tails (including curvature in notochord and spinal cord and spiraling of the tail; see Fig. 4d, *yellow arrows*, for a representative profile). Strikingly, these RS-induced effects only occurred in animals developed without brain. The RS treatment of Ctrl animals (normal development with brain, BR⁺) had no effect on tail patterning, and severe phenotypes were not detected. The analysis of the frequency of distribution of the different phenotypes within each population revealed significant differences among the experimental conditions (Fig. 4e; $X^2_{(0.05,6)} = 370.8$; $P < 0.01$). The macroscopically identifiable changes in tail patterning, induced by RS treatment in BR⁻ embryos, were also accompanied by clear qualitative alterations in the fine muscle structure and somite organization (Fig. 4f, g). We conclude that presence of the brain helps embryogenesis resist the disrupting effects of otherwise strongly teratogenic agents.

To identify possible pathways mediating this brain-protecting effect, we tested whether scopolamine (which rescues BR⁻-muscle phenotype) could also protect against the effects of teratogens in BR⁻ embryos. We found that in the absence of a brain, scopolamine treatment partially counteracted the teratogenic effects of RS (Scopo⁺, Supplementary Fig. 3A–C). Scopolamine-treated RS-BR⁻ embryos (Drug⁺BR⁻Scopo⁺) are much healthier than BR⁻ embryos, but they do not exhibit an entirely recovered Ctrl-like muscle phenotype (Supplementary Fig. 3D). Macroscopic tail phenotype and muscle structure analysis revealed that scopolamine significantly decreases the occurrence of highly aberrant phenotypes (from $92 \pm 2\%$ in Drug⁺BR⁻Scopo⁻ to $15 \pm 1\%$ in Drug⁺BR⁻Scopo⁺, $X^2_{(0.05, 4)} = 136.3$; $P < 0.01$), leading to a phenotype distribution similar to what occurs in the regular BR⁻ population ($31 \pm 9\%$ in BR⁻ without drug treatment or Drug⁻BR⁻Scopo⁻). We conclude that scopolamine treatment prevents the severe deformities caused by this teratogen, but its effect is not sufficient to fully prevent the muscle defects caused in BR⁻ by the presence of the drug.

**The absence of brain leads to abnormal neural development**. Having seen the profound effects of the brain on the developing musculature, we next asked what type of alterations/reorganizations could have occurred on the remaining nervous tissue after brain removal (peripheral innervation).

We visualized the body-wide neural network at late-stage embryos by immunolabeling them with acetylated alpha-Tubulin antibody (Tub) and quantifying the Tub-immunolocalization by OD measurements (ranging from 0 (black, no expression) to 255 (white, maximal expression)). This antibody is a widely-accepted marker for nerve fibers because stabilized microtubules, such as those found in neuronal processes, contain important amounts of acetylated tubulin[39]. The typical neural pattern in the peripheral nervous system (PNS) of Ctrl embryos revealed that, at late stages, three types of fibers can be clearly identifiable after Tub-immunostaining (Fig. 5a): (i) commissural nerve fibers, running along dorsoventral axis (long arrows); (ii) longitudinal nerve fibers, along anteroposterior axis (short arrows); and (iii) internal network or neural network underlying the space

between two consecutive segments defined by the commissural ones (unfilled triangles). The internal neuropil in Ctrl animals consisted of a thin network, barely detected by OD measurements (OD mean value of $15 \pm 7$ units). After brain removal, and similarly to errors detected for segmentation, commissural and longitudinal fibers in BR⁻ were mispatterned (note some incorrect commissural nerve distribution coincident with defects at the level of the somitic myogenesis, magenta arrow in Fig. 5b). Strikingly, embryos developed without a brain exhibited a robust ectopic branching (internal neuropil), with nerve fibers chaotically orientated through the animal body (yellow unfilled triangles in Fig. 5b, OD mean value of $32 \pm 11$ units; $P < 0.01$ compared to Ctrl group, after post-hoc Bonferroni's test). Results from sham-surgery embryos (extirpating non-brain regions) confirmed that this nerve misspatterning is specifically due to brain removal, as demonstrated after Tub immunolocalization (Supplementary Fig. 4). We conclude that the absence of a brain provokes a dramatic and specific change of the branching of the internal nerve net in the whole animal body (aberrant neural network).

To test whether the effect of brain removal was due to lack of an endogenous pruning phase, we analyzed the motoneuron axonal patterning in early stage embryos (stages 31–41; Supplementary Fig. 5) using the antibody znp1. This antibody labels primary motoneuron axons[40], and is widely used for many different animal models during embryogenesis[26]. We found no significant difference in the density of motor axonal branches (internal neuropil) among stage 31, 35 and 45 Ctrl embryos (after post-hoc Bonferroni's test). Conversely, both errors in axonal establishment (magenta arrows in Supplementary Fig. 5B, D, F showing lack of trajectory compared to turquoise arrows in Supplementary Fig. 5A, C, E) and ectopic/aberrant branching (yellow unfilled triangles in Supplementary Fig. 5B, D, F pointing branches projecting off the main axons; compare to turquoise unfilled triangles in Supplementary Fig. 5A, C, E) were detected from the onset and during the progression of the development in BR⁻ embryos. Thus we conclude that the brainless phenotype involved ectopic growth of neural tissue, not a failure of normal pruning.

Having detected that ectopic branching was a BR⁻-induced specific effect on peripheral nerve structure, we asked next if the pharmacological treatment and/or ectopic ion channel expression used to rescue the muscle phenotype would have similar effects on the BR⁻-aberrant neural network (Fig. 5c–f). Results derived from cholinergic-drug treatment revealed that the BR⁻-induced dense internal neuropil was not fixed by scopolamine or carbachol. Unlike the rescue effects on muscle phenotype, scopolamine was not able to prevent the massive internal neural branching (Fig. 5c; OD mean value of $31 \pm 5$ units; $P < 0.05$ compared to Ctrl group, after post-hoc Bonferroni's test). Carbachol-treated BR⁻ embryos showed a completely disorganized nerve structure, more aberrant than those in drug-untreated BR⁻, with aberrations in the three different types of nerve fibers (Fig. 5d; OD mean value for internal neuropil of $37 \pm 9$ units; $P < 0.01$ compared to Ctrl group, after post-hoc Bonferroni's test). However, analysis of HCN2-injected embryos showed that the nerve sprouting consequent to developing without a brain was efficiently rescued by expression of HCN2 WT (Fig. 5e; OD mean value of $25 \pm 11$ units; $P > 0.05$ compared to Ctrl group, after post-hoc Bonferroni's test). We conclude that the aberrant neural network in BR⁻ can be fixed by ion channel misexpression providing additional channels, but not by the pharmacological treatment targeting existing ones.

Given that the BR⁻-induced aberrant neural network was rescued by HCN2 overexpression, we tested the spatial signaling between HCN2-expressing cells and the responding PNS, by

means of quantitative evaluation of the nerve patterning in the contralateral uninjected side of 1/2 HCN2-WT overexpressing embryos (see text above and Fig. 3b for experimental injection details). 65% of uninjected sides (low HCN2 expression) had internal neuropil similar to those in both injected side (high HCN2 expression), and Ctrl animals (compare internal neuropil in a typical contralateral uninjected side (Fig. 5g) to that one in injected side (Fig. 5h) and Ctrl animal (Fig. 5a)), displaying an OD mean value of 20 ± 5 units (P > 0.05 compared to Ctrl group, after post-hoc Bonferroni's test; the quantitative evaluation of the intensity of Tub protein signal on injected side was not performed because OD values could not be comparable to the other analysis, due to the characteristic dark precipitate in the cells expressing β-gal). Our results suggest that the rescue effects of HCN2 are not only mediated by the cells expressing this specific channel (autonomous cell behavior), but that, in absence of brain, the alteration of bioelectrical state could promote accurate nerve patterning via long-distance signals.

**Brain regulates muscle and nerve patterning via distinct modes.** Next we investigated the possible pathway by which brain acts on distant tissues: electrical efferent pathway, via spinal cord, vs. alternative or exo-spinal pathway, by severing the spinal connection between brain and the rest of the body (Fig. 6a and Table 3). We analyzed the muscle and nerve patterning in embryos developed with a brain, but with a cervical fragment of spinal cord resected at stage 25 (SC⁻; Fig. 6a top panel). The mean

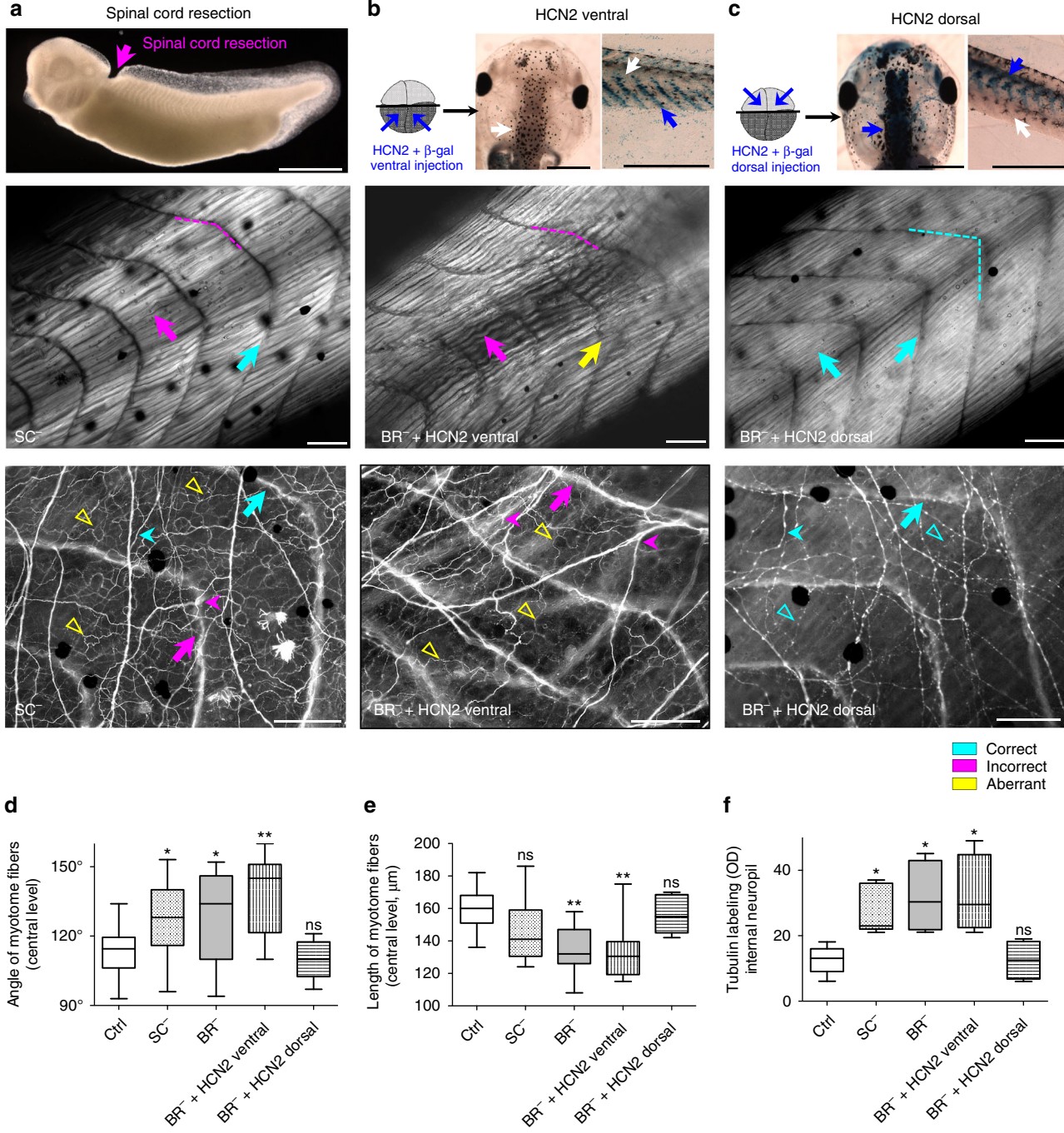

angle of the myotome fibers in SC$^-$ ($128 \pm 16°$) differed significantly from the typical chevron-shape angle in Ctrl animals ($114 \pm 10°$, $P < 0.05$ after post-hoc Dunn's test), leading to an overall altered muscle organization (Fig. 6a middle panel, magenta dashed arrowhead-like line). However, SC$^-$ presented less severe muscle phenotype than BR$^-$: most of the myofibers presented normal fine structure (with non-significant differences in the mean length of the myotome fibers compared to Ctrl, after post-hoc Bonferroni's test) with unfrequented structural aberrations. SC$^-$ neural patterning exhibited some degree of organization for commissural and longitudinal fibers, but frequent errors were present (magenta arrows in Fig. 6a bottom panel). Internal neuropil was, nevertheless, profoundly altered (yellow unfilled triangles), displaying the typical BR$^-$ aberrant or ectopic nerve branching. We conclude that while muscles can develop moderately well without direct spinal cord-dependent brain signaling, the observed brain effects on nerve patterning require an intact spinal cord.

Having seen the differential effect of brain signaling on muscle and nerves, and considering the HCN2 rescue effects, we next specifically targeted the dorsal (neural) regions or ventral (somatic muscle) regions of brainless animals with HCN2 mRNA. Embryos were microinjected with HCN2 (wild-type channel) and lacZ mRNA either in the two ventral cells (Fig. 6b upper panel) or in the two dorsal cells (Fig. 6c upper panel) at the four-cell stage. Embryos with HCN2 ventral injections that developed without a brain (BR$^-$ + HCN2 ventral) presented profound defects in muscle structure, both in angle and in length/organization of the myotome fibers (Fig. 6b middle panel). Ectopic or aberrant patterning was also present, as seen in BR$^-$. Conversely, embryos with HCN2 dorsal injections that developed without a brain (BR$^-$ + HCN2 dorsal) presented a perfectly organized myotome, with normal myofiber structure and organization, indistinguishable from typical Ctrl-muscle patterning (Fig. 6c middle panel). Nerve patterning in BR$^-$ + HCN2 ventral animals was markedly altered for all the different fiber types, with an extensive and mispatterned intermyotomal nerve branching (Fig. 6b lower panel). Conversely, HCN2-mRNA injections in dorsal module of the embryo lead to a well-organized nerve phenotype, indistinguishable from what occurs in Ctrl embryos (Fig. 6c lower panel). Quantification is shown in Fig. 6d–f. We conclude that in order to rescue the BR$^-$-induced effects, HCN2 needs to be expressed in dorsal structures (neural tube).

## Discussion

Here we show that the early morphogenesis and patterning of trunk muscle structure and innervation in animals developing without a brain are highly abnormal. Brainless (BR$^-$) animals' peripheral neural network is profoundly disorganized, with fibers chaotically oriented through the animal body, while the muscle organization was adversely affected both at the microscopic tissue organization level (length/definition and angle of myotomes), as well as at the animal morphological level (aberrant phenotype). The effect is brain-specific, as removal of other body regions does not induce this effect. The brain is not only required for normal development, but also exerts a protective effect, ameliorating the effects of teratogenic drugs which are made notably worse in brainless embryos.

We gained insight into the mechanism of brain-dependent, long-range patterning effects by rescue assays. Ectopic expression of a hyperpolarization-activated cyclic nucleotide-gated ion channel (HCN2) was sufficient to prevent muscle and nerve mispatterning in brainless animals. The HCN2 rescue effect only occurs when CNS-fated blastomeres are targeted, suggesting that bioelectrical signals, when acting within neural tissues, can mimic the endogenous effects of the brain. Future work will determine the relative contributions, to the HCN2 rescue, of modulating spiking-encoded activity in the nervous system[41], and alteration of non-neural distributions of resting potential that have likewise been implicated in developmental patterning[42].

We also started exploring the potential therapeutic implications of our findings, by recapitulating the protective effects using pharmacological agents targeting endogenous channels (not requiring exogenous misexpression). Drugs targeting the cholinergic system differentially affected BR$^-$-induced outcomes. A dual nicotinic and muscarinic agonist exacerbated the defects in muscle structure; in contrast, suppression of muscarinic pathway, by means of scopolamine treatment, rescued it. The fact that scopolamine can partially rescue the defects in muscle, but not the aberrant nerve phenotype, and that spinal cord-transected animals develop a partially normal muscle phenotype, suggest that the brain could play a direct role in muscle development that may not involve spinal pathway and peripheral nerves (Fig. 7a, b). Taken together, these data reveal an essential role for brain-derived signaling during embryogenesis, long before its involvement in behavior, and show that the patterning effects of the

**Fig. 6** Brain effects on muscle and nerve patterning are partially mediated via spinal cord and mimicked via the dorsal expression of HCN2. **a–c** Upper row, **a** Lateral view of a stage-33 embryo following spinal cord resection (SC$^-$) at stage 25. Site of injury is indicated by magenta arrow. **b**, **c** Embryos were microinjected with HCN2 (wild-type channel) and lacZ mRNA either in the two ventral cells (**b**, blue arrows) or two dorsal cells (**c**, blue arrows) at the four-cell stage. Animals were evaluated at stages 42–48. HCN2-ventral embryos were β-galactosidase negative (β-gal$^-$, white arrow) for brain (center image in **b**, dorsal view) and SC (right image in **b**, lateral view) and β-gal$^+$ (blue arrow) for ventral myotomes (right image in **b**). HCN2-dorsal embryos were β-gal$^+$ for brain (center image in **c**) and SC (right image in **c**) and β-gal$^-$ for ventral myotomes (right image in **c**). For lateral views, rostral is left and dorsal is up. Scale bar, 500 μm. Middle row, Typical muscle phenotype for SC$^-$ (left panel), HCN2-ventral injected BR$^-$ (center panel), and HCN2-dorsal injected BR$^-$ (right panel), as seen under polarized light. Muscle patterning (angle of the myotomes, magenta dashed arrowhead-like line) in SC$^-$ was altered compared to Ctrl. SC$^-$ presented a less severe phenotype than BR$^-$ displaying myofibers with normal structure (turquoise arrow) and some incorrect patterning (magenta arrow). BR$^-$ + HCN2 ventral embryos presented profound defects in muscle structure, both in angle (magenta dashed line) and in length/organization (magenta arrow) of the myotome fibers. Ectopic or aberrant patterning was also present (yellow arrow). BR$^-$ + HCN2 dorsal embryos presented an organized myotome, with normal myofiber structure and organization (turquoise dashed line and arrows). Lower row, typical nerve patterning (commissural fibers indicated by long turquoise arrow, longitudinal fibers indicated by head arrows, and internal neuropil indicated by unfilled triangles) for SC$^-$ (left panel), HCN2-ventral injected BR$^-$ (center panel) and HCN2-dorsal injected BR$^-$ (right panel), shown on anti-acetylated alpha-tubulin antibody staining. SC$^-$ exhibited some degree of organization for commissural and longitudinal fibers (turquoise arrows), but frequent errors were present (magenta arrows). Internal neuropil was, nevertheless, profoundly altered, displaying the typical BR$^-$aberrant or ectopic nerve branching (yellow). Nerve patterning in BR$^-$ + HCN2 ventral was markedly altered for all the different fiber types. Conversely, HCN2-mRNA injections in dorsal cells lead to an entirely well-organized nerve phenotype, indistinguishable from controls. Rostral is upper right and dorsal is up. Scale bar, 100 μm. **d–f.** Quantification of the mean angle (**d** Kruskal–Wallis, $P < 0.01$) and length (**e** one-way ANOVA, $P < 0.01$) of central myotome fibers and Tub-positive internal neuropil (**f** one-way ANOVA, $P < 0.01$), along with statistical comparisons for each experimental group vs. Ctrl ($P$ values above the bar). Data represent the mean OD units and s.d. of two independent replicates ($n = 50$ animals per group). $P$ values after post-hoc analysis are indicated as **$P < 0.01$, *$P < 0.05$, ns $P > 0.05$

**Table 3 Muscle and nerve measurements for Ctrl, BR⁻, SC⁻, BR⁻ with ventral HCN2 injections, and BR⁻ with dorsal HCN2 injections**

|  | Muscle | | Nerve |
|---|---|---|---|
|  | Angle (°) | Length (μm) | Branching (OD units) |
| Ctrl | 114 ± 11° | 159 ± 13 | 13 ± 4 |
| BR⁻ | 115 ± 12°* | 136 ± 14** | 32 ± 11* |
| SC⁻ | 128 ± 16°* | 145 ± 17ⁿˢ | 28 ± 7* |
| BR⁻ + HCN2 ventral | 138 ± 17°** | 132 ± 17** | 32 ± 12* |
| BR⁻ + HCN2 dorsal | 109 ± 8ⁿˢ | 156 ± 11ⁿˢ | 12 ± 6ⁿˢ |

Ctrl: control; BR⁻: brainless; SC⁻: spinal cord resected; OD: optical density
Values are presented as mean and s.d. Statistical comparisons to Ctrl group are indicated for each group. P values are indicated as **$P < 0.01$, *$P < 0.05$, ns $P > 0.05$ (*black labels* after post-hoc Bonferroni's and *blue labels* after post-hoc Dunn's test)

brain can be largely mimicked by available reagents targeting cells' bioelectric state.

*X. laevis* is uniquely suited for the study of biophysical mechanisms underlying pattern regulation. Somites in *X. laevis* are comprised of myotome fibers[43] and embryonic myogenesis in *Xenopus* involves intricate interplays between several MRFs: MyoD, Myf5, Myf6 (also called Mrf4), Myogenin and Myf6. Dynamic temporal and spatial expression patterns of these factors orchestrate the main steps of muscle development: lineage specification of muscle cells, differentiation of myocytes, fusion into myofibers, and formation of muscle groups (reviewed in ref. [23]). Future developments integrating in vivo physiological monitoring with transcriptional reporters will address the interaction of neurotransmitter and bioelectric signals from the brain with the transcriptional control of these and other important factors.

Our results show that brain input is important for patterning and myotome organization (Fig. 1c–h) rather than the early events of myogenesis, such as fate or induction of cell lineages. Somite segmentation is not altered by brain removal, as demonstrated by the normal number of somites in BR⁻ animals at late stages, suggesting that the periodicity of somite formation and initiation of myogenesis (mediated mainly by MyoD and Myf5) do not require early brain-derived signaling. Brain inputs might be acting in later events, when the myotome fibers have been already established in the somite. At these differentiation steps, expression of Mrf4 and Myogenenin might be susceptible to brain signaling. In *Xenopus*, Mrf4 has been showed to be the main myogenic factor subject to nerve influence. This evidence is, however, conflicting and some authors postulate that the initial activation of Mrf4 is nerve-independent in embryonic *Xenopus*[44]. However, different studies demonstrate a neural influence on Mrf4 expression. In fact, muscle denervation leads to decrease Mrf4 levels, in both development and regeneration[19]. Each myogenic factor might have a distinct role in the regulation of nerve-regulated genes, such as different subunits of a neuromuscular junction (NMJ) receptor: the nicotinic acetylcholine receptor nAChR[45].

A variety of studies have demonstrated the relationship between innervation and correct anatomical development of muscle structures, in different vertebrates. Denervation of rat skeletal muscles in utero provokes degeneration and myofiber fragmentation, as well as a slowing down of myofiber growth[46]. Similar findings were seen in frogs, where the denervation of the hind limb leads to a ~ 12% reduction in growth, and for the development of limb transplants in chick embryos[47]. Abnormalities in overall patterning (size and shape) of the limbs in absence of nerve influence have been also described in salamanders, being the muscle the most sensitive to nerve absence[48].

Such data are usually thought to be explained by 'trophic' or permissive effect of nerves[49]. Our results, where brainless embryos have more innervation than the control ones (Fig. 5a, b), suggest an additional and 'instructive' role, mediated by signals normally originating in the brain.

Our results reveal a long-distance role of brain-derived signaling at both organ level (overall muscle phenotype; Fig. 2a, b) and tissue level (myofibrillar structure; Fig. 2g, h). The partially fixed muscle phenotype after the spinal cord resection or after scopolamine treatment seems to indicate that in addition to the spinal pathway, muscle is susceptible to a long-distance action of the brain, perhaps via diffusion of neurotransmitter signals (schematized in Fig. 7).

Some of the most tantalizing data reveal a role for the brain and CNS in patterning of distal structures. Older studies have postulated a role of the brain-derived signals, which are conducted along the spinal cord, on morphogenesis in *Xenopus* tail regeneration; the subcommissural organ was suggested as the source of this signaling[50]. Recent experiments using point ablation in the spinal cord[14] showed that patterning of the final regenerated tail is influenced for both the injury position along the AP axis and a non-linear combinatory effect when two different AP injury levels are performed. Hence, shape-instructive long-distance signaling is not explained by simple presence of nerve (trophic effects) but appears to generate distinct information along different positions in the spinal cord. Our 'hypercurvature' phenotype in BR⁻ animals (Fig. 4e) is comparable to Mondia et al.'s most severe one, suggesting that similar signals from brain might are acting on development and regeneration.

We report the first steps to identify the mediators of the early brain's effect by showing that the effects on myopatterning derived from the absence of a brain can be completely rescued by ectopic alteration in neurotransmitter (Fig. 2) and ion channel (Figs. 3 and 6c middle panel) signaling.

Even in the absence of a brain, muscarinic AChR (mAChR) suppression (via scopolamine treatment) led to close-to-normal muscle development, while nicotinic AChR (nAChR) and mAChR activation (carbachol treatment) provoked a more aberrant muscle phenotype. Exogenous application of acetylcholine modulates the intrinsic properties of spinal motoneurons after SC transection in the juvenile salamander through the mAChR[51]. Likewise, functional nAChR have been reported in spinal motoneurones of *X. laevis* embryo[52]. These studies support the reorganization in spinal cord circuits in absence of brain as the main target for actions of scopolamine and carbachol on muscle cells. The alteration of membrane potential ($V_{mem}$), by a direct action of these drugs on receptors in the early muscle cells, could be an important part of the mechanism. We hypothesize that scopolamine is acting on NMJ, specifically at presynaptic/synaptic level, blocking the Ach actions by mAChRs on slow ion flows[52] ($I_f$; Fig. 7d). This $I_f$ disinhibition could counter the excess of excitability induced by the presence of ectopic branching in BR⁻. Our results suggest a strategy to pharmacologically target muscular defects, by means of regulating the balance between slow $I_f$ and action potentials at NMJ level.

Prior work revealed patterning roles of several neurotransmitters[37, 53], and suggested that neurotransmitter drugs could be potent teratogens[36]. We found that an NMDA agonist provoked severe phenotypes in BR⁻ animals (RS, Fig. 3). NMDA-glutamate receptors (NMDAR) in muscles are also excitatory and depolarize the muscle cells leading to their contraction (such as carbachol does on nAChR). Given recent work on the importance of steady-state developmental voltage gradients during *Xenopus* muscle patterning[31], our findings are consistent with the hypothesis that maintaining the balance between developmental bioelectricity (resting potential gradients) and discrete action

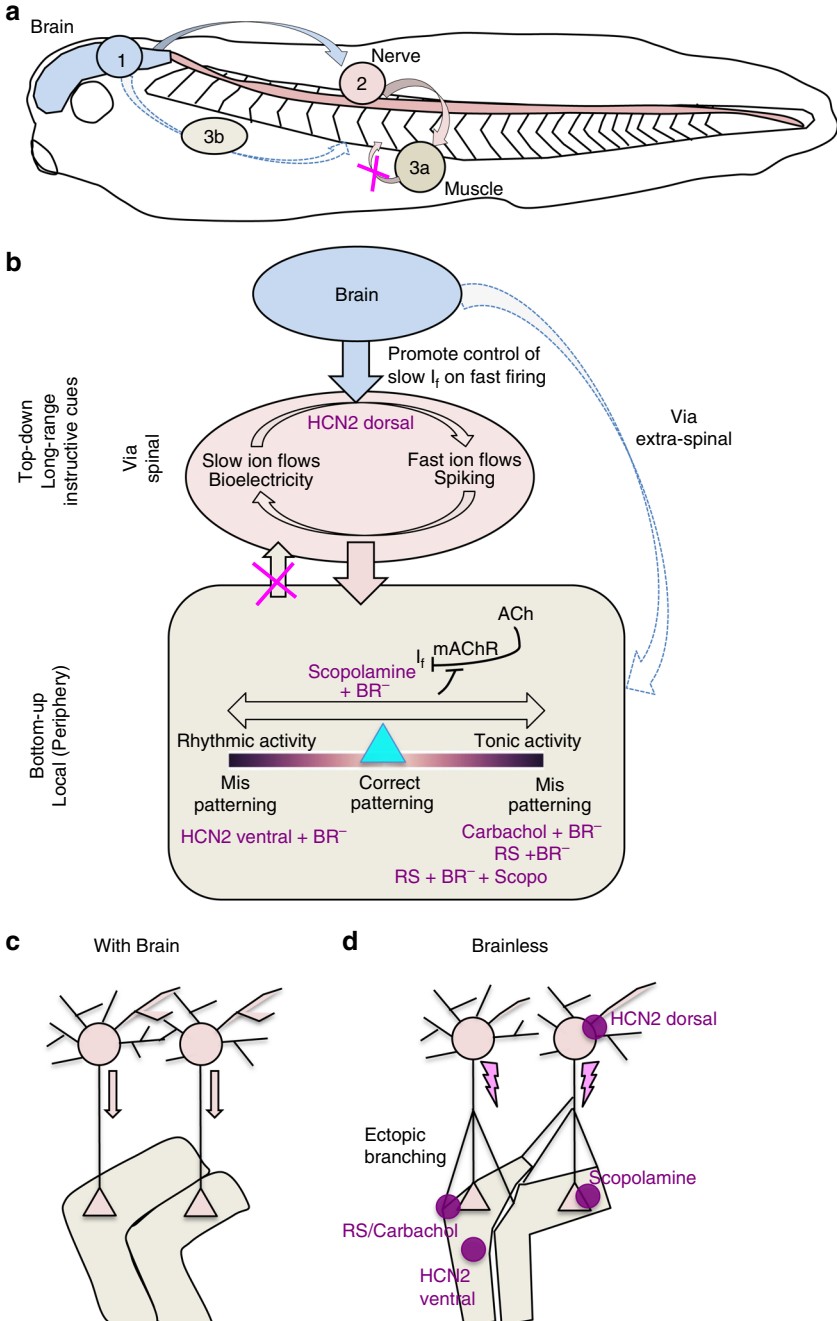

**Fig. 7** Brain signaling for muscle and nerve development and patterning **a**. Schematic representation drawing of a *Xenopus* embryo, showing the main components of our experiments: brain (*blue*), spinal cord-peripheral nerves (*pink*) and somites-muscle (*brown*). Brain effects on nerve patterning could occur directly (2), by using efferent spinal pathway. Brain effects on muscle patterning could occur indirectly (3a), by acting on neurons, or directly (3b), by acting on muscle. **b** A spinal mechanism, for coding the information about patterning and morphogenesis, could occur via direct signaling from the brain to the neurons in the spinal cord (*pink circle*). According to our results (different treatments are indicated with *purple labels*), the effects of the peripheral innervation on muscle cells can be partially explained in terms of developmental bioelectricity or changes in $V_{mem}$ excitability. We hypothesize that at these stages, the brain is in part controlling the bioelectric state of peripheral tissues, and a correct balance (*turquoise triangle*) of brain activity (long-range instructive cues or top-down perspective) and local signals (bottom-up perspective) is necessary for correct morphogenesis. Both an excess of tonic activity (induced after carbachol or RS treatment) and an excess of slow $I_f$ gradients through membrane lead to mispatterning. The extra-spinal pathway by which the brain is acting on muscles can be mimicked pharmacologically, with pharmacological agents targeting bioelectricity (i.e., scopolamine). We hypothesize that scopolamine is acting at presynaptic/synaptic level, blocking the inhibitory ACh actions (via mAChRs) on slow ion flows, and leading the $V_{mem}$ to appropriate values for muscle patterning. **c**, **d**. Schematic representation of neuromuscular specificity in normal development (**c**, with brain) and in absence of the brain (**d**, BR⁻). Our results suggest that ectopic branching detected in the absence of a brain is not due to deficits in early pruning or target retrograde signaling. Pathfinding behavior at the onset of *Xenopus* development starts at the spinal cord level, as early patterned electrical gradients in SC cells is required for the correct axon guidance. The different treatments applied in our experiments (*purple labels* and *circles*) are placed on the cellular/subcellular domains where they are probably acting

potentials at NMJ may be important during muscle development and inhibitory signals from the brain (probably via extra-spinal alternative pathways) may be acting to shield the developing muscle cells from such excitatory stimulus that might cause muscle mispatterning (Fig. 7b, brown rectangle representing muscle membrane). Recently, it has been showed that activation of NMDAR impairs the myogenic differentiation in C2C12 cells through mTOR/MAPK signaling pathway[54]. The protective effect of brain detected in the RS-treated control animals (vs. the devastating effects observed in RS-treated BR⁻-animals) evidences the key role of the brain signaling for the correct morphogenesis. The possibility of exploiting and perhaps strengthening brain-derived protective signals represent an exciting area of research for future efforts in the field of birth defects.

In addition to the muscle effects, the absence of a brain during development generates an abnormal patterning and organization of the peripheral innervation or PNS of the animal (Fig. 5; somatic component of the PNS). We used immunohistochemistry with an antibody to acetylated α-tubulin[39] to analyze the somatic neural processes coming from both primary motoneurons and sensory neurons, in order to study the instructive role of brain inputs in patterning and global organization of the nervous system.

Cell fate and differentiation for primary motoneurons and sensory neurons start early in embryogenesis. Rohon-Beard (RB) neurons cells originate during gastrulation and present electrical excitability as early as stage 20[24]. Neural crest (NC) material segregates at gastrula stage (around stage 15) and starts migrating around stage 20[25]. In our assay, brain removal is done after the onset of the migration, at stage 25, and hence, when the fate of the trunk NC cells has been already specified[55].

In neural morphogenesis, after differentiation and migration, and once the progenitors have reached the final location, new steps are necessary for axon growth and guidance (pathway selection), formation of initial connections (target selection), and connection remodeling and pruning (address selection). The early neural morphogenesis, differentiation and migration of X. laevis, both for sensory and motor somatic neurons[56], earlier than the time point for the brain removal, lead us to suggest that aberrations detected in neural patterning in BR⁻ animals might be due, to the later steps on pathfinding and synapse formation. Thus, the peripheral innervation pattern is an epigenetic outcome that depends on complete development of the functional physiology of the brain; in brain removal, not all of the signals required for a correct peripheral pattern are conveyed.

The peripheral innervation formed in brainless animals is mispatterned throughout the whole animal body (Fig. 5). How neuromuscular specificity arises during embryonic development has been a controversial issue. Are correct connections pre-established from the outset or do motor axons project randomly into the developing muscles followed by extensive pruning of incorrect connections? Our analysis of brainless animals (Supplementary Fig. 5) shows that nerves are altered very early in development (as soon as stage 31), with clear errors in finding the correct trajectory (actually in BR⁻ nerve fibers fail in turning to create the correct intermyotome division, see Supplementary Fig. 5B, ventral magenta arrow). Our results suggest that brain-derived signals are important for correct early patterning, not for maintenance of a default pruning program.

HCN2 targeted to dorsal part (brain and spinal cord) rescues both muscle and nerve. Spinal cord-transected embryos show the same aberrant ectopic branching than BR⁻, suggest that pathfinding behavior at the onset of Xenopus development starts at the spinal cord (or CNS) level (Fig. 7c; as previously was showed in zebrafish[57] or axolotl[58]). Moreover, in line with recent evidence[59], we show that the early patterned electrical in spinal motoneurons (as HCN2 dorsal is able to rescue the whole phenotype and brain effects on nerve patterning require an intact spinal cord) is required for the correct axon guidance. While previous work did not explain the mechanisms by which the rhythmic activity in spinal neurons affects early nerve development, our data are consistent with a role for the brain in determining patterned morphogenesis by controlling slow ion flows on primary-neuron fast firing (Fig. 7b).

Interestingly, we discovered a rescue effect on the neural phenotype only by means of reagents that target membrane potential ($V_{mem}$), induced by the ectopic expression of HCN2 (Figs. 5e–h and 6c bottom panel). While targeted misexpression of ion channels has been demonstrated to rescue patterning of the brain itself[32], we report here that using ion channel expression can overcome developmental defects stemming from brain damage. Future work will exploit the emerging advances in optical imaging of neural activity in vivo[60] to characterize the temporal properties of brain-derived signals and their modification by HCN2.

One remarkable aspect is that HCN2 expression can rescue patterning of cells on the other side of the animal – cells that do not themselves express HCN2 (Fig. 5g). Such long-range bioelectrical signaling has been observed in bioelectric tumor suppression[61] and control of apoptosis/proliferation[32]. Our data reveal that expression of HCN2 at a remote location can induce repair of peripheral neural structures in a damaged background, suggesting a range of therapies where easily-accessible tissues are targeted to induce repair in a difficult-to-reach site. Future work extending closed-loop optogenetic control to neural and non-neural somatic tissues[62] will refine the specific bioelectric state that facilitates normal network structure and test these interventions in adult disease models.

Scopolamine-treated BR⁻ animals display an aberrant neural network but normal muscle phenotype, suggesting distinct regulatory mechanisms. We do not claim that correction in nerve patterning (by HCN2 injection; Fig. 5e–h) is the only factor responsible for muscle correction by that treatment (Fig. 3b). The direct action of ion channels in muscle cells and their indirect action on muscle patterning through peripheral nerves are not mutually exclusive possibilities. Future work will be directed towards understanding their mutual contributions to the repair process. The fact that pharmacological modulation of Ach transmission does not affect the neural patterning (Fig. 5c, d) but bioelectrical modulations do (as in vitro studies seems also to indicate[63]), suggests that both long-distance brain input and developmental bioelectricity are potential targets for future applications in the area of muscle-nerve communication pathologies.

Here, we establish an experimental model for the study of long-range patterning control, and the discovery of pre-behavior functions of the nascent brain. This model is amenable to optical, biophysical, genetic, and chemical approaches, and offers the unique opportunity to target diverse spatial sites (due to the Xenopus fate-map) to test non-cell-autonomous mechanisms of brain-dependent instructive patterning signals. Future development of transgenic promoters will allow tissue-specific tests, while targeted ablation technology may enable finer dissection of brain regions responsible for various patterning outcomes.

Our data suggest a revision of the view of the brain as quiescent prior to the animal's independent activity, showing that its signaling role spans the control of pattern formation and behavior. This is consistent with recent proposals that the important mechanistic and conceptual commonalities exist between the algorithms of neuroscience and those that guide pattern regulation[64]. This model system and dataset serves as a base for future studies of local and long-range influences over

large-scale patterning. The relationship of instructive morphogenetic signals mediated by bioelectric events to the computational capabilities of the brain is an exciting direction for future work. Moreover, our data point to widely-available and already human-approved drugs as potential 'morphoceuticals' – agents that can be capitalized upon to prevent or perhaps even reverse specific kinds of anatomical defects. More broadly, these results suggest a direction for regenerative medicine towards the development of implanted organoids and hybrid electrochemical constructs to provide bioelectric and neurotransmitter stimulation.

## Methods

**Animal husbandry**. *Xenopus laevis* embryos were fertilized *in vitro* according to standard protocols[25] in 0.1X Marc's Modified Ringer's solution (MMR; 10 mM Na$^+$, 0.2 mM K$^+$, 10.5 mM Cl$^-$, 0.2 mM Ca$^{2+}$, pH 7.8). *Xenopus* embryos were housed at 14 °C and staged according to Nieuwkoop and Faber[65]. All experimental procedures involving *Xenopus* embryos were approved by the Institutional Animal Care and Use Committees and Tufts University Department of Laboratory Animal Medicine under protocol M2014-79.

**Microsurgery**. Stage-25 embryos were randomly grouped in one of the two experimental groups: Control (Ctrl) and Brainless (BR$^-$; see Fig. 1a for a schematic representation of the experimental design). Randomized controlled trial conditions were maintained throughout the experiment. Embryos were anesthetized in a 0.02% tricaine solution (pH 7.5) in 0.1X MMR. Brain removal in BR$^-$ group was performed under dissecting microscope and using a dissecting knife (FST #10055-12). Once movement ceased, a single cut removed the anterodorsal region corresponding to the brain (from the cement gland to the most anterior somite; Fig. 1b). After brain removal, animals were allowed to heal in 0.75X MMR for 1 h. After washing, untreated and surgically treated animals were raised at 14 °C and scored and analyzed for phenotype distribution and the different morphological parameters, respectively, at stages 30–41 (early stage) and 42–48 (late stage). Following the same experimental and care conditions, additional spinal cord (SC) resection (SC$^-$; Fig. 6a) experiments were performed on stage-25 embryos. To prevent regeneration[66], one segment of the spinal cord, sizing 50–100 μm in length, was completely removed by using two forceps with super-fine tips (Dumont #5ST, FST 11252-00). SC segments were removed at cervical level, immediately posterior to the hindbrain, and taking special care of minimizing the damage on the most rostral myotomes.

Two different Sham surgeries were performed by removing pieces of tissue of comparable size at different locations of the stage-25 animal body (Supplementary Fig. 1): part of the endodermal yolk mass (Yolk$^-$ or yolk resection) or the most posterior part of the embryo body (Tail$^-$ or tailbud resection).

In order to feed the BR$^-$ tadpoles, a nutritive medium was modified from ref. [67]. The nutritive medium consisted of 9.5% Ham's nutrient mixture F12 (with 1.0 mM L-glutamine, Sigma 51651 C) and 0.5% calf serum (from formula-fed bovine calves, iron supplemented, Sigma C8056) dissolved in 0.1X MMR. From stage 45, Ctrl and BR$^-$ tadpoles were fed with this nutritive medium for 2 h every day.

**Microinjections**. Capped synthetic mRNAs generated using mMessage mMachine kit (Ambion) were dissolved in nuclease-free water and injected into embryos immersed in 3% Ficoll solution using standard methods[25]. The mRNA injections were made using borosilicate glass needles calibrated to bubble pressures of 55 to 60 kPa in water, delivering 100- to 130-ms pulses. Each injection delivered between 0.5–1 nl or 0.5–1 ng of mRNA per blastomere into the embryos. Two different sets of injections were performed. Firstly, at the two-cell stage (Fig. 3a, b), either one (1/2 HCN2 Inj) or both (HCN2 Inj) blastomeres were injected. Secondly, at the four-cell stage (Fig. 6b, c), either the two ventral (HCN2 ventral Inj) or the two dorsal (HCN2 dorsal Inj) blastomeres were injected. Injections were done into the center of cells at the animal pole.

Constructs used were HCN2-WT-2A-GFP or HCN2-WT[68] and β-galactosidase in PCS2 and both RNAs (HCN2-WT and β-galactosidase) were mixed at 3:5 and 1:5 dilutions respectively, for microinjections. HCN2 is a potassium/sodium hyperpolarization-activated cyclic nucleotide-gated ion channel, type 2[69]. Embryos were injected in 3% Ficoll solution and after 30 min, were washed and then reared in 0.1X MMR until desired stages.

**Drug exposure**. Ctrl and BR$^-$ *Xenopus* embryos were exposed to specific pharmacological agents, dissolved in 0.1X MMR, from stage 25 (immediately after brain removal) to stage 48. The drugs were refreshed every three days. Two drugs targeting cholinergic receptors and one directed to NMDA-glutamatergic receptor were used: 10 μM scopolamine (a muscarinic-receptor antagonist; Tocris 1414), 10 μM carbamoylcholine chloride (carbachol, a dual muscarinic- and nicotinic-receptor agonist; Tocris 2810) and 10 μM (*RS*)-(Tetrazol-5-yl)glycine (RS; an NMDA receptor agonist; Tocris 0312), respectively. All drug treatments were

performed using embryos from mixed batches of fertilizations, using at least three biological replicates.

Stock solutions of all pharmaceuticals were created by dissolving the compound in Millipore water (to a final drug concentration of 100 mM for both scopolamine and carbachol, and to 50 mM for RS) and then stored in aliquots at –20 °C. Further dilution of all compounds was made in normal media (0.1X MMR). Control experiments were performed using embryos in 0.1X MMR, both for Ctrl and BR$^-$ groups. Drug concentrations were determined through toxicity screens and were applied at levels that did not result in lethality or observable developmental defects.

**Immunofluorescence and histochemistry**. Immediately after treatment, at the relevant time points (immediately after brain removal and first 2 weeks and third and fourth weeks after brain removal for early- and late-stage studies, respectively), anesthetized tadpoles were fixed in MEMFA at 4 °C for whole-mount immunofluorescence[25]. Briefly, embryos were washed twice in 1× phosphate buffered saline (PBS), and permeabilized in PBS 0.1% Triton X-100 (PBST) for 30 min. Animals were then blocked with 10% normal goat serum in PBST for 1 h at room temperature (RT). Samples were rocked overnight at 4 °C using znp1 (Developmental Studies Hybridoma Bank, used at 1:250 dilution) and anti-acetylated alpha-tubulin antibody (Tub; Sigma T7451 used at 1:500 dilution). Following primary exposure, embryos were washed three times in PBST before a 60-min RT incubation with AlexaFluor-555 conjugated secondary antibody (Invitrogen) used at 1:500 diluted in PBST. Following secondary incubation, animals were washed three times for 15 min in PBST and imaged on an Olympus BX-61 microscope equipped with a Hamamatsu ORCA AG CCD camera, and controlled by Metamorph software. Particular care was taken to ensure that embryos from all the different groups were processed in the same batch at the same immunofluorescence session.

**Beta-galactosidase enzymatic detection**. Embryos injected with β-galactosidase (β-gal) mRNA were fixed (30 min in MEMFA at RT) at the relevant stages, washed twice in PBS with 2 mM MgCl$_2$, and stained with X-gal (Roche Applied Sciences, Indianapolis, IN) staining solution at 37 °C for 3 h. Embryos were then rinsed three times in PBS and analyzed.

**In situ hybridization**. *Xenopus* embryos at stage 35 were collected and fixed in MEMFA[25] and in situ hybridization was performed[25, 70]. The embryos were washed with phosphate buffered saline 0.1% Tween-20 (PBST) and transferred through a series of methanol washes 25%-50%-75%-100%. In situ antisense probe was generated in vitro from linearized template using DIG labeling mix (Roche). Chromogenic reaction times were optimized for signal to background ratio. Antisense RNA probes for *Xenopus* HCN2 were generated from *X. laevis* hcn2.L IMAGE clone 5514485 (purchased from Dharmacon): a HindIII fragment was deleted, leaving exons 2-4 and part of exon 5 as probe. In situ hybridized embryos were then agarose embedded and sectioned. Briefly, 4% low melting point agarose was melted at poured into plastic scaffolds. Gently dried, in situ hybridized embryos were oriented within the agarose for transverse sectioning. The agarose blocks were then allowed to solidify, trimmed and sectioned using Leica VT1000S vibratome to obtain transverse sections.

**Phenotype scoring and morphological evaluation**. Ctrl-, Sham-Yolk$^-$-, and BR$^-$-*Xenopus* embryos were scored for abnormalities in the macroscopic phenotype. Average number of animals with clearly identifiable left-right axis bending and/or tail defects were used to evaluate the percent abnormal within each population. Embryos were photographed (lateral view) using a Nikon AZ100 with an attached QImaging CD camera controlled by QCapture software.

Mean collagen density, angle and length of myotome fibers, and number of somites. All muscle structure studies were performed on 4× or 10× images taken under polarized light. Birefringence microscopy was performed on an Olympus BX-61 compound microscope with a universal condenser (U-UCD8). The transmitted light DIC slider (U-DICTS) was pulled out and the polarizing filter was rotated such the background appeared darkest. Embryos were positioned at a 45° angle for imaging. Analysis of birefringence images was performed in ImageJ (National Institutes of Health, Maryland, US).

Collagen density in early-staged embryos (Fig. 1c, d; *short arrows*) was evaluated by means of optical density (OD) using a gray scale gradient of 0-255 (white to black), on 4× polarized-light photomicrographs[31]. Each embryo was measured using a systematic procedure in order to sample the OD at different anatomical levels, along the anteroposterior axis. Six OD-myotome measurements were used to determine mean value for each animal. At least 30 embryos within each experimental group were analyzed. Possible aberrations originating from the optical system of the camera were corrected by background image subtraction.

Mean angle and length of myotome fibers (Fig. 1c–g; overlaid *dashed arrowhead-like lines* and *double-headed arrows*, respectively) for both early- and late-staged embryos was performed on 10× polarized-light images, using ImageJ software. For each animal, three somites at anterior, central and posterior levels, respectively, were systematically random sampled. To obtain a representative mean value of length, nine myotome fibers, from dorsal to ventral axis, were measured for each somite at each anatomical level. The number of somites, between end of

the gut and tip of the tail, for both early- and late-staged embryos was evaluated on 4x birefringence images.

Analysis of the neural network branching (see Fig. 5, *unfilled triangles*) in early- and late-staged embryos was performed by evaluating the intensity of znp1 and acetylated alpha-tubulin (Tub) immunostaining, respectively, through gray-level measures (OD). In order to obtain one OD-mean value per animal, multiple measurements were taken along the anteroposterior axis. Each measurement consisted of the mean value of the pixels of a fixed-size window, placed between two consecutive somites (intermyotomal or internal neuropil). The size of the window remained constant across subjects. All individuals among whom comparisons are being made were produced in the same batch, treated identically for processing and imaging–conditions were not changed. In addition, possible aberrations originating from the optical system of the camera were corrected by background image subtraction.

**Statistics**. All statistical analysis was performed using GraphPad Prism (GraphPad Software, Inc., CA, US) and Microsoft Excel (Microsoft Corporation, WA, US) software. Each dish of tadpoles was considered a replicate.

When comparing numerical variables (OD, number of somites, angle and length of the myotome fibers), data from, at least, two replicates (minimal replicate size: $n = 25$ embryos) were analyzed per each experimental condition. Firstly, data were tested for homogeneity of variances by Bartlett's test. If variances are similar, we applied unpaired and two-tailed Student's $t$-test (two independent groups), one-way ANOVA test (multiple independent groups), or two-way ANOVA test (two independent variables) followed by post-hoc Bonferroni's test (when $P < 0.05$). In case of unequal variances (or non-normal distributions for a sample size of less than 50), two-tailed Mann–Whitney and Kruskal–Wallis test followed by pot-hoc Dunn's test (when $P < 0.05$), respectively, were used.

For differences in sample proportion (percent of aberrant embryos) or distribution analysis (frequency of phenotypes), data from, at least, two replicates and sample size equal or greater than 50 embryos were pooled and analyzed by multiple-sample $z$-test or $X^2$ test, respectively.

The significance level ($\alpha$) was set to 0.05 in all cases. The statistical values are reported as mean ± s.d.

**Data availability**. The authors declare that all data supporting the findings of this study are available within the article and its Supplementary Information Files or from the corresponding author upon reasonable request.

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

## Acknowledgements

We thank Rakela Colon and Erin Switzer for general laboratory assistance and *Xenopus* husbandry, Dany S. Adams for help with microscopy, Nian-Qing Shi for the HCN2-WT DNA construct and Jean-Francois Pare for assistance with the sub-cloning of cDNA HCN2 into PCS2. We thank Kelly McLaughlin and Joshua Finkelstein for their helpful comments on the manuscript. This paper is dedicated to the memory of Panagiotis 'Takis' Tsonis, a profound multi-disciplinary scientist who significantly impacted our understanding of biological control systems. This research was supported by the Allen Discovery Center program through The Paul G. Allen Frontiers Group (12171). We also gratefully acknowledge support from the W.M. Keck Foundation (5903), the G. Harold and Leila Y. Mathers Charitable Foundation (TFU141), and the National Institutes of Health (AR055993, AR061988).

## Author contributions

C.H.-R. performed experiments (microsurgery, developmental analysis, drug experiments, immunofluorescence and birefringence imaging). M.L. and C.H.-R. designed the experiments and interpreted data. V.P.P. performed mRNA injection experiments. K.M.M. assisted with the immunofluorescence experiments and the spinal cord experiments. J.M.L. cloned the HCN2 mRNA construct that was used for the in situ experiments. C.H.-R., M.L. and V.P.P. wrote the paper together.

## Additional information

**Competing interests:** : The authors declare no competing financial interests.

