## [Peer Review File · Nature Communications]

Reviewers' Comments:

Reviewer #1 (Remarks to the Author)

This paper investigates a novel idea--that the brain has a role in the patterning of trunk/tail muscles and peripheral nerve network in developing *Xenopus*. The authors assay the effect of amputating the brain at the start of somitogenesis, then describe the use of a pharmacological or overexpression approach to investigate the mechanism of the effect. They show that the effect of brain amputation on both muscle and nerve patterning is profound—somite segmentation and number is not affected but the overall morphology of the embryo and the structure of muscle are distorted, and the peripheral neural network pattern is hyper-developed and disorganized. The effects on each appear largely unrelated to the other.

The muscle defects, but not the neural defects, can be rescued by the pharmacological inhibition of muscarinic Ach receptors by scopolamine, suggesting that this kind of inhibition is required for normal development and is a function of brain signals. Both the muscle and the peripheral nerve pattern were rescued by overexpression of the hyperpolarizing HCN2 ion channel, which plays a pacemaker role in the heart and in the sensation of pain via nociceptive neurons. When expressed on only one side of the BR- embryo, the rescue effect on muscle pattern is observed on the other side as well, indicating communication between the two sides. They also found by this approach that the brain protects muscle and neural pattern from damage inflicted by teratogenic agents.

The results of this investigation add, in an original way, to the growing body of evidence that the developing nervous system has a major role in morphogenesis and patterning, in this case at least via neurotransmitters and ion channel polarization. In particular, the results reveal a long-distance role of brain signals superimposed on local signals.

The investigation is well-conceived, interesting, original and the data back up the conclusions. Some comments and critiques are as follows:

P. 6, line 132—specify “Soon” (St. 30-41?)

Line 147—specify stage (42-48?)

P. 9, line 231. A clearer statement would be “...we conclude that the brain inhibits the muscarinic pathway...”

P. 10, lines 259-260. Please make sure to state that injection was in both blastomeres, to distinguish from the one blastomere injection.

What is the pattern of HCN2 expression in the normal CNS over the time points of these experiments, and is expression lost in BR- embryos?

Can scopolamine or overexpression of the HCN2 channel protect against the effects of teratogens (RS) in BR- embryos?

What are the long-distance signals produced by alteration of the bioelectric state mediated by HCN2?

The discussion is very long and rich, and suggests a number of potential translational aspects of this research for the future, particularly for the HCN2 channel results. These are already being explored for therapeutic intervention in neuropathic pain (Tsantoulas et al (2016) *Biochem J.* 473 (18): 2717-36. I was particularly intrigued by the part of the discussion from lines 572-597. The

authors appear to say that neural defects in BR- embryos are not the result of interference with early neural morphogenesis, but on later steps in pathfinding and synaptogenesis on the periphery. The peripheral pattern does not develop completely autonomously, but appears to be an epigenetic function that depends on complete development of the brain pattern. With brain removal, not all the epigenetic signals for a correct peripheral pattern can be conveyed. Is this a correct interpretation?

Overall, a novel and valuable piece of work.

David L. Stocum

Reviewer #2 (Remarks to the Author)

In this manuscript, Michael Levin and colleagues present first evidence for the involvement of the brain in muscle and nerve patterning during early development in *Xenopus*. The authors developed a method to ablate the brains of living embryos, exposed embryos to acetylcholine receptor agonists and antagonists, and overexpressed HCN2 channels to (partly) rescue the observed phenotype with disturbed muscle and nerve patterning. Additionally, they find that brain tissue protects the developing embryo from teratogens.

The authors thus identify a new, exciting function of the brain in regulating morphogenesis, which should be highly interesting to a wide readership. The experiments are thoroughly done and well described (although the manuscript and particularly the discussion would benefit from shortening parts; it is currently a bit lengthy and not as concise as it could be), the figures are excellent, and the data showing that the brain somehow controls muscle and nerve patterning in the periphery are convincing.

However, I have two concerns that should be addressed before I can recommend publication.

First, the authors need to motivate much better why they have chosen the perturbations described in the manuscript. Why are acetylcholine receptors interesting candidates to begin with? Currently, the pharmacological experiments using agonists and antagonists come out of the blue, and I had a hard time finding the rationale for these experiments.

Second, there is currently little mechanistic insight into how the brain might regulate muscle and nerve patterning. The authors are rather vague and repeatedly talk about brain-derived signals, which, however, could be electrical activity, hormonal signals, or other signalling molecules (either directly or indirectly impacting muscle development) etc. It is also not clear if in brainless animals peripheral nerves or muscles are primary sites of the effect, and if one is following the other, or if both systems are independently hit by the missing signal(s) from the brain. The authors state 'we conclude that the brain may use the muscarinic pathway to achieve correct organization of the somatic muscle system' – how does it do this, how is this supposed to work? This is not clear to me. A schematic figure summarizing the results of this study might help.

In my opinion, the most likely scenario that might explain all experimental findings is that nerve endings at neuromuscular junctions are established but not pruned in the absence of the brain (as activity is essential for pruning), hence there are many more nerves than in control animals, and an AChR antagonist and membrane hyperpolarization both downregulate the activity of neuromuscular junctions. This would, however, contradict the conclusion that 'The rescue effects of scopolamine on muscle but not on nerve structure suggest a direct role of brain-derived signals on muscle development, rather than an indirect effect acting via alterations in the peripheral nerves.', which is not really backed up by any data. This should be experimentally tested, for example by looking at nerve patterns at an earlier time point, before the onset of pruning. If brainless and control animals have similar phenotypes at such an earlier stage, this would hint towards this mechanism (which is actually briefly discussed at the end of the manuscript).

Also, to dissect whether the effect of brain removal is primarily on neurons or on muscles, one of the cell types should be targeted specifically (currently all perturbations hit both neurons and muscle cells, as well as other cell types in the tissue). For example, HCN2 channels were expressed in all cells. To disentangle the contribution of nerves and muscles, I would recommend to inject the plasmids at a later time point, for example at the 4 or 8 cell stage, into CNS-fated blastomeres. This way, only neurons but not muscle cells would be hyperpolarized.

To test if the effect of the brain is via electrical or chemical signalling, a control could be done in which the brain is left in the embryos but the spinal cord is locally cut and/or partially removed (to avoid regeneration), or literature could be cited. If the 'brain-derived signals' are electrical, this treatment would eliminate them, while chemical signals would persist in the embryo.

Minor points:

1. The clock wavefront model mentioned in the introduction is not accurately describing somite development. I'd suggest to include a reference to Soroldoni et al., Science, 2014.
2. The language when comparing relative changes between groups should be changed, e.g., 'with a percentage change of up to 25%' should rather be stated, for example, as 'with a 25% decrease in length'; 'respect to' should be replaced by 'compared to', 'reaching up to 10% lower of mean length' by 'whose length was decreased by 10%', etc.
3. 8bit corresponds to 256 grey levels (2^8), so grey values should range from 0 to 255 (not from 1). Were the measurements done in some ratiometric way to normalize for general differences in staining and noise between the groups, bleaching etc.? Or have the parameters of the image acquisition at least not changed between groups? This should be explicitly stated in the text.
4. Page 13: 'After brain removal, and according to alterations detected for somitogenesis, the distribution of commissural and longitudinal fibers in BR- seemed to follow the pattern established by the segmentation' – couldn't it also be the other way around, i.e., the segmentation patterning following the fiber patterns?
5. Page 14: 'We conclude that the aberrant neural network in BR- can be only fixed by ion-channel misexpression' – I find this very unlikely. There are probably other, currently unexplored ways of achieving a similar effect?
6. Many conclusions (and particularly the medical relevance) should generally be toned down a bit. Another example from page 16: 'Brainless (BR-) animals' peripheral neural network is completely disorganized' – completely is a very strong word.
7. Page 20: 'The rescue effect of HCN2 (a hyperpolarizing channel; Fig. 3B-3G) is consistent with the hypothesis that keeping Vmem hyperpolarized prevents the muscle damage in BR - animals.' – Do the authors suggest that the role of the brain is to hyperpolarize axons and render them inactive?

Reviewer #1

The results of this investigation add, in an original way, to the growing body of evidence that the developing nervous system has a major role in morphogenesis and patterning, in this case at least via neurotransmitters and ion channel polarization. In particular, the results reveal a long-distance role of brain signals superimposed on local signals. The investigation is well-conceived, interesting, original and the data back up the conclusions. Some comments and critiques are as follows:

We thank the referee for their positive assessment and constructive comments. We have made improvements as follows:

P. 6, line 132—specify “Soon” (St. 30-41?)

Line 147—specify stage (42-48?)

Done.

P. 9, line 231. A clearer statement would be “...we conclude that the brain inhibits the muscarinic pathway...”

Done.

P. 10, lines 259-260. Please make sure to state that injection was in both blastomeres, to distinguish from the one blastomere injection.

Done.

What is the pattern of HCN2 expression in the normal CNS over the time points of these experiments, and is expression lost in BR- embryos?

Done. We have included the *in situ* hybridization data in this version of the manuscript (page 10, lines 264-266), as well as in a new figure (Supplementary Figure 2). We performed the *in situ* hybridization analysis of endogenous *HCN2* expression in Control and BR⁻ embryos at stage 25, 35 and 45 (Supplementary Figure 2) and found that at all three stages, the *HCN2* channel is expressed in the developing neural tube particularly in the basal and lateral regions within the developing neural tube (Supplementary Figure 2). It is also present in the perisomic area (Supplementary Figure 2). In BR⁻ embryos, the *HCN2* expression pattern remains largely unchanged.

Can scopolamine or overexpression of the HCN2 channel protect against the effects of teratogens (RS) in BR- embryos?

This is a really interesting question. The HCN2 experiments will be performed as part of very extensive on-going studies focused on this channel, as it has some unique patterning and signaling properties which space does not permit to delve

into in this paper; these will be reported in a forthcoming manuscript. We focused our new experiments on obtaining the answer to the reviewer's question regarding scopolamine. We found that scopolamine treatment indeed prevents drug-induced teratogen effects on BRainless (BR⁻) embryos (Supplementary Figure 3). Macroscopic tail phenotype and muscle structure analysis revealed that scopolamine countered the effects of RS, decreasing significantly the occurrence of highly aberrant phenotypes and creating a phenotype distribution similar to the displayed in the regular BR⁻ population (BR⁻ without drug treatment or Drug-BR⁻ Scopo⁻) population ($X^2_{(0.05, 4)}=136.3$; $P<0.01$). We have included these experiments in this version of the manuscript (page 13, lines 342-356), as well as in a new figure (Supplementary Figure 3).

What are the long-distance signals produced by alteration of the bioelectric state mediated by HCN2?

This is an excellent question, as a next research program emerging from this paper. We have begun a study of the non-local effects, but this is a difficult and long-term line of investigation. At the moment, we do not know what mediates the long-range signaling. Possibilities include flows (either direct or relay) of specific ions (especially calcium), current, or neurotransmitters. These are difficult to verify, because they are very small molecules and cannot be easily imaged. We are developing functional and biochemistry assays to try to reveal them. We look forward to reporting our results on this in a subsequent publication.

The discussion is very long and rich, and suggests a number of potential translational aspects of this research for the future, particularly for the HCN2 channel results. These are already being explored for therapeutic intervention in neuropathic pain (Tsantoulas et al (2016) *Biochem J.* 473 (18): 2717-36. I was particularly intrigued by the part of the discussion from lines 572-597. The authors appear to say that neural defects in BR-embryos are not the result of interference with early neural morphogenesis, but on later steps in pathfinding and synaptogenesis on the periphery. The peripheral pattern does not develop completely autonomously, but appears to be an epigenetic function that depends on complete development of the brain pattern. With brain removal, not all the epigenetic signals for a correct peripheral pattern can be conveyed. Is this a correct interpretation? Overall, a novel and valuable piece of work.

Thank you, this is an excellent point. Pathfinding and synaptogenesis require not only regulation of gene activity for protein expression, subcellular packing and microtubule motors, but also all the epigenetic events such as brain activity and neural communication across distance. According to our results, brain signals are mostly affecting the multi-scale pathway/target/address-selection processes (and not the earlier cell fate and differentiation), because they must be orchestrating across a number of subcellular compartments, including temporal and spatial control of gene expression, as well as cell behaviors. Emphasizing this in the context of epigenetic guidance and control is a brilliant point which we have now

incorporated into the paper. We have now revised our text to be clearer on this point (page 23, lines 647-650).

Reviewer #2

The authors thus identify a new, exciting function of the brain in regulating morphogenesis, which should be highly interesting to a wide readership. The experiments are thoroughly done and well described (although the manuscript and particularly the discussion would benefit from shortening parts; it is currently a bit lengthy and not as concise as it could be), the figures are excellent, and the data showing that the brain somehow controls muscle and nerve patterning in the periphery are convincing.

We thank the referee for their positive and detailed assessment. We have attended to all of the points he/she raised as follows:

First, the authors need to motivate much better why they have chosen the perturbations described in the manuscript. Why are acetylcholine receptors interesting candidates to begin with? Currently, the pharmacological experiments using agonists and antagonists come out of the blue, and I had a hard time finding the rationale for these experiments.

Done. We have now added more text explaining the rationale for the choice of reagents in the Result section (page 8, lines 188-199), as follows:

“Neurotransmitters, such as acetylcholine, are conserved and ubiquitous mediators of the brain’s electrical activity on other organs and tissues in the body. We reasoned that similar mechanisms might be at work prior to behavior, in the developmental process, as neurotransmitters not only mediate adult physiological function downstream of bioelectrical events but also play a developmental role in the patterning and formation of the synapses that they subserve^{45,46}. Therefore, we targeted this pathway to attempt to understand and recapitulate the brain’s role in embryogenesis. We tested several cholinergic drugs that were known to target muscarinic (mAChRs) and nicotinic cholinergic receptors (nAChRs) and are standard tools for altering brain performance, especially in terms of memory, attention, and relevant-stimulus processing⁴⁷.”

Second, there is currently little mechanistic insight into how the brain might regulate muscle and nerve patterning. The authors are rather vague and repeatedly talk about brain-derived signals, which, however, could be electrical activity, hormonal signals, or other signalling molecules (either directly or indirectly impacting muscle development) etc.

It is true that at this point, we do not know all of the signals. However, our functional data are remarkable in that they identify interventions that are *sufficient* to rescue absence of brain. Thus, we reveal at least some of the molecular nature of these signals: for example acetylcholine, which is a central player and hard to classify as distinctly chemical or electrical as it mediates both (neurotransmitters, especially ones which gate ion channels, in developmental contexts are a nexus at which the terms “electrical” and “chemical” lose their sharp distinction). We hypothesize that the full symphony of signals are a sequential mix of electrical and

neurotransmitter-mediated events. Because tracking the individual flows of ions and neurotransmitters is not really feasible *in vivo* with today's technology (either in the live state or in immunohistochemistry – they are too small to be labeled), we are working on understanding this long-range signaling as part of the next-stage of this work. Specifically, we are collaborating with chemists to develop fluorescent and enzymatic sensors, as well as functional (optogenetic) approaches, that could be used to dissect this signaling, but it may be a year or more before such technology becomes available and effective. We look forward to presenting those data as part of a forthcoming submission.

It is also not clear if in brainless animals peripheral nerves or muscles are primary sites of the effect, and if one is following the other, or if both systems are independently hit by the missing signal(s) from the brain.

We have now clarified this in the text and illustrated the signaling pathways in the new Figure 7. Our data – that Scopolamine treatment rescues muscle but not nerves – indicates that the two systems are not inextricably coupled, because we can rescue one of them without altering the other. Thus, there is a degree of independence, although HCN2 expression in dorsal cells rescues both nerves and muscle. It can be hypothesized that rescuing nerves is enough to have normal muscles, and we now discuss this possibility on pages 23-24 (lines 663-677). Future experiments using (not yet available) transgenic frog lines, which will allow independent targeting of reagents to subpopulations of nerve and muscle precursors, will allow this question to be settled with greater cell-type resolution in subsequent work.

The authors state 'we conclude that the brain may use the muscarinic pathway to achieve correct organization of the somatic muscle system' – how does it do this, how is this supposed to work? This is not clear to me. A schematic figure summarizing the results of this study might help.

Done. We have now provided a schematic model comprising and illustrating all of our data, as suggested (new Fig. 7).

In my opinion, the most likely scenario that might explain all experimental findings is that nerve endings at neuromuscular junctions are established but not pruned in the absence of the brain (as activity is essential for pruning), hence there are many more nerves than in control animals, and an AChR antagonist and membrane hyperpolarization both downregulate the activity of neuromuscular junctions. This would, however, contradict the conclusion that 'The rescue effects of scopolamine on muscle but not on nerve structure suggest a direct role of brain-derived signals on muscle development, rather than an indirect effect acting via alterations in the peripheral nerves.', which is not really backed up by any data.

We apologize for lack of clarity on this point, we have now revised the text (pages 18-19, lines 511-515). Our sentence was not meant to convey that brain only (or mainly) acts on muscle. What we meant is simply that brain signals (in this case,

chemical signals) can act on muscle independently on what happens on peripheral nerves. So, peripheral nerves and muscle can be *independently* regulated by signaling from the brain.

This should be experimentally tested, for example by looking at nerve patterns at an earlier time point, before the onset of pruning. If brainless and control animals have similar phenotypes at such an earlier stage, this would hint towards this mechanism (which is actually briefly discussed at the end of the manuscript).

This is a very interesting suggestion, and we have now performed new experiments to address it (Supplementary Figure 4). Our results after immunofluorescence with znp1 antibody (specific marker for primary motoneuron axons^{2,5-7}) show that the aberrant neural branching that occurs in the absence of a brain in BR⁻ embryos is *not* due to deficits in early pruning. Control animals showed no evidence of significant pruning effect on the early-established motor axons (Supplementary Figure 4 A, C, E, G). These results are in the line with previous studies reporting that such pruning takes place in *Xenopus* at stages much later than the ones we describe^{8,9}. Conversely, ectopic branching was already detected in BR⁻ embryos at the earliest stage 31 and it was present throughout subsequent development (Supplementary Figure 4 B, D, F, G), as well as at the later stages (as shown in Fig. 5 of the previous version of the manuscript).

The new results have been included in the updated version of the manuscript, in both Results (pages 14-15, lines 390-404) and Discussion sections (page 23, lines 652-662), as well as in a new figure about it (Supplementary Figure 4). A schematic model has been also provided in Fig. 7C, D.

Parenthetically, we used znp1 antibody (instead of using anti-acetylated alpha tubulin, like in our experiments of the first version) because at these earlier stages, the high density of Tub⁺ ciliated (non-neuronal) cells (as shown below, white arrows) makes almost impossible to consistently and unambiguously identify and analyze the Tub⁺ nerves (blue arrowhead-like unfilled triangles), compared to the clear znp1 motor axonal labeling, see Supplementary Figure 4). See the following page:

Low-magnification (A) and high-magnification (B) photomicrographs of stage-35 Control embryos after immunofluorescence with anti-acetylated alpha tubulin (Tub) showing both Tub-positive elements: ciliated epidermal cells (white arrows) and axons (blue arrowhead-like unfilled triangles). The high density of ciliated cells, typical of these early developmental stages, makes difficult the analysis of the nerve patterning. Scale bar = 100 μm . Rostral is lower right, dorsal is up.

Also, to dissect whether the effect of brain removal is primarily on neurons or on muscles, one of the cell types should be targeted specifically (currently all perturbations hit both neurons and muscle cells, as well as other cell types in the tissue). For example, HCN2 channels were expressed in all cells. To disentangle the contribution of nerves and muscles, I would recommend to inject the plasmids at a later time point, for example at the 4 or 8 cell stage, into CNS-fated blastomeres. This way, only neurons but not muscle cells would be hyperpolarized.

Done. We have now performed these experiments. **We found that the effect of brain removal is primarily on neurons**, as HCN2 rescue effect is only detected when dorsal cells at four-cell stage embryos are injected and ectopically express the channel. Ventral ectopic expression of HCN2 in BR⁻ does not rescue muscle or nerves, indicating that only when neural plate cells are provided with the hyperpolarizing HCN2 ion flux, muscle can be entirely rescued. These new data have been included in the updated version of the manuscript, in both Results (page 17, lines 463-481) and Discussion sections (pages 23-24, lines 663-677), as well as in a new figure about it (Fig. 6, B and C modules). A schematic model has been also provided in Fig. 7C, D.

To test if the effect of the brain is via electrical or chemical signaling, a control could be done in which the brain is left in the embryos but the spinal cord is locally cut and/or partially removed (to avoid regeneration), or literature could be cited. If the 'brain-derived signals' are electrical, this treatment would eliminate them, while chemical signals would persist in the embryo.

Done. This suggestion was a great way to address the role of spinal cord in brain-derived signals' propagation. We have now performed this experiment. We found that brain effects on muscle and nerve patterning are partially mediated via spinal cord (Fig. 6A-A"), as embryos developed with brain but with spinal cord transected (SC-) presented alterations in both muscle and nerve patterning. These SC-induced alterations are *partial* because muscle phenotype is not as severely affected as in BR-, showing even non-significant differences for length of the myotome fibers when compared to Ctrl embryos. This might indicate that brain control on muscle development is through two parallel pathways: indirect (via spinal cord, acting on nerves, as having correct nerves is sufficient to have correct muscle), and direct via extra-spinal (as muscles in SC- and scopolamine-treated BR- show some degree of organization, even if nerves are severely affected).

These new data have been included in the updated version of the manuscript, in both Results (pages 16-17, lines 444-462) and Discussion sections (pages 22-23, lines 663-677), as well as in a new figure about it (Fig. 6A). A schematic model has been also provided in Fig. 7A, B.

Minor points:

1. The clock wavefront model mentioned in the introduction is not accurately describing somite development. I'd suggest to include a reference to Soroldoni et al., Science, 2014.

Done.

2. The language when comparing relative changes between groups should be changed, e.g., 'with a percentage change of up to 25%' should rather be stated, for example, as 'with a 25% decrease in length'; 'respect to' should be replaced by 'compared to', 'reaching up to 10% lower of mean length' by 'whose length was decreased by 10%', etc.

Done.

3. 8bit corresponds to 256 grey levels (2^8), so grey values should range from 0 to 255 (not from 1).

We apologize for this mistake in explaining the methodology that was actually used. We have now revised the text (page 14, lines 365-366; page 31, lines 874-875), including the correct range. We note that this does not change our results, as the analysis was done using ImageJ which scales the values correctly (0-255) – it was only our methods text that stated the wrong minimum number.

Were the measurements done in some ratiometric way to normalize for general differences in staining and noise between the groups, bleaching etc.? Or have the parameters of the

image acquisition at least not changed between groups? This should be explicitly stated in the text.

Done. We now state in the text (page 32, lines 899-903) that all individuals among whom comparisons are being made were produced in the same batch, treated identically for processing and imaging – conditions were not changed. In addition, possible aberrations originating from the optical system of the camera were corrected by background image subtraction.

4. Page 13: ‘After brain removal, and according to alterations detected for somitogenesis, the distribution of commissural and longitudinal fibers in BR- seemed to follow the pattern established by the segmentation’ – couldn’t it also be the other way around, i.e., the segmentation patterning following the fiber patterns?

Yes, this is possible and we have now revised our text accordingly. The sentence has been changed to: “After brain removal, and similarly to errors detected for segmentation, commissural and longitudinal fibers in BR- were miss patterned.” We intentionally avoided mentioning here any temporal sequence because the discussion has been extensively updated in this regard.

5. Page 14: ‘We conclude that the aberrant neural network in BR- can be only fixed by ion-channel misexpression’ – I find this very unlikely. There are probably other, currently unexplored ways of achieving a similar effect?

Fixed. We did not mean to imply that no other treatment could ever be found that has this effect, simply that among the treatments we attempted, the channel misexpression worked but the pharmacological reagents did not. The sentence has been changed to: “We conclude that the aberrant neural network in BR- can be fixed by ion-channel misexpression but not by the pharmacological treatments applied in our experiments.”

6. Many conclusions (and particularly the medical relevance) should generally be toned down a bit. Another example from page 16: ‘Brainless (BR-) animals’ peripheral neural network is completely disorganized’ – completely is a very strong word.

Done. We have now corrected this and other instances. The sentence in question has been changed to: “Brainless (BR-) animals’ peripheral neural network is profoundly disorganized.”

7. Page 20: ‘The rescue effect of HCN2 (a hyperpolarizing channel; Fig. 3B-3G) is consistent with the hypothesis that keeping Vmem hyperpolarized prevents the muscle damage in BR

– animals.’ – Do the authors suggest that the role of the brain is to hyperpolarize axons and render them inactive?

This is an excellent question; indeed a study in zebrafish concluded that the abnormal axonal phenotypes may be related to a lack of depolarizing activity early in development⁶. However, one key issue has to be kept in mind: in developmental bioelectricity (unlike in pure nerve/muscle physiology), there is not a binary dichotomy between active and inactive. During development, a wide range of V_{mem} patterns appears and changes over timescales much longer than millisecond spiking (classical “activity”). We (and others) have shown in many papers that these gradients are an important endogenous component of correct pattern formation. Thus, we now make it clearer in the text that we hypothesize that at these stages, the brain is in part controlling the bioelectric state of peripheral tissues, and a correct balance of brain activity and signals is necessary for correct morphogenesis. HCN2 is a hyperpolarizing channel that is subtle – it reacts to surrounding conditions (cAMP levels, for example) and can even work as a pacemaker in stem cells. We think that by expressing this channel, we are helping to restore the correct balance of V_{mem} throughout target tissues, during this developmental transition between pre-neural somatic bioelectric patterns (analog, slow) and mature spiking dynamics in nerve and muscle (discrete, rapid). We now mention all this in the text, to be clear on this point. A schematic model has been also provided in Fig. 7B.

References cited:

- 1 An, M. C. *et al.* Acetylcholine negatively regulates development of the neuromuscular junction through distinct cellular mechanisms. *Proc Natl Acad Sci U S A* **107**, 10702-10707, doi:10.1073/pnas.1004956107 (2010).
- 2 Behra, M. *et al.* Acetylcholinesterase is required for neuronal and muscular development in the zebrafish embryo. *Nature neuroscience* **5**, 111-118, doi:10.1038/nn788 (2002).
- 3 Owens, J. L. & Kullberg, R. Expression of nicotinic acetylcholine receptors in aneural *Xenopus* embryos. *Dev Biol* **135**, 12-19 (1989).
- 4 Klinkenberg, I. & Blokland, A. The validity of scopolamine as a pharmacological model for cognitive impairment: a review of animal behavioral studies. *Neuroscience and biobehavioral reviews* **34**, 1307-1350, doi:10.1016/j.neubiorev.2010.04.001 (2010).
- 5 Adams, K. L., Rousso, D. L., Umbach, J. A. & Novitch, B. G. Foxp1-mediated programming of limb-innervating motor neurons from mouse and human embryonic stem cells. *Nature communications* **6**, 6778, doi:10.1038/ncomms7778 (2015).
- 6 Menelaou, E. *et al.* Embryonic motor activity and implications for regulating motoneuron axonal pathfinding in zebrafish. *The European journal of neuroscience* **28**, 1080-1096, doi:10.1111/j.1460-9568.2008.06418.x (2008).
- 7 Trevarrow, B., Marks, D. L. & Kimmel, C. B. Organization of hindbrain segments in the zebrafish embryo. *Neuron* **4**, 669-679 (1990).
- 8 Lamb, A. H. Neuronal death in the development of the somatotopic projections of the ventral horn in *Xenopus*. *Brain Res* **134**, 145-150 (1977).
- 9 Herrera, A. A. & Werle, M. J. Mechanisms of elimination, remodeling, and competition at frog neuromuscular junctions. *Journal of neurobiology* **21**, 73-98, doi:10.1002/neu.480210106 (1990).

Reviewers' Comments:

Reviewer #1:

Remarks to the Author:

The authors have addressed all of the critiques of the reviewers, including the addition of new experiments. I find the MS to be very strong; thank you for your attention to the details.

Reviewer #2:

Remarks to the Author:

The authors have put a lot of effort into the revised version of the manuscript and have addressed all reviewer comments. The manuscript is now a very convincing piece of work that will undoubtedly be highly interesting to a wide readership.